# SSFO: Self-Supervised Faithfulness Optimization for Retrieval-Augmented Generation

## Abstract

Retrieval-Augmented Generation (RAG) systems require Large Language Models (LLMs) to generate responses that are faithful to the retrieved context. However, faithfulness hallucination remains a critical challenge, as existing methods often require costly supervision and post-training, or imposing significant inference burdens. To overcome these limitations, we introduce Self-Supervised Faithfulness Optimization (SSFO), a self-supervised alignment approach for enhancing faithfulness. SSFO constructs preference data pairs by contrasting the model's outputs generated with context versus without context. Leveraging Direct Preference Optimization (DPO), SSFO aligns model faithfulness without incurring labeling costs or additional inference burdens. We analyze this faithfulness alignment process and provide empirical evidence that it leverages a benign form of *likelihood displacement*, shifting probability mass from parametric-based tokens to context-aligned tokens. Based on this insight, we adapt the DPO loss using a weighting scheme that encourages likelihood displacement. Comprehensive evaluations show that SSFO significantly outperforms existing methods, achieving state-of-the-art results in faithfulness on multiple context-based question-answering datasets. Notably, SSFO exhibits strong generalization, improving cross-lingual faithfulness while preserving general instruction-following capabilities. The code is available at: `https://anonymous.4open.science/r/SSFO`

## 1 Introduction

With the widespread deployment of Retrieval Augmented Generation (RAG) (Lewis et al., 2020; Jokinen, 2024), Large Language Models (LLMs) (Achiam et al., 2023; Touvron et al., 2023) are increasingly expected to generate responses that adhere closely to the provided context (Song et al., 2025; 2024; Niu et al., 2024). However, an LLM's parametric knowledge from pre-training can interfere with the provided context and lead the model to generate unsupported information, known as *faithfulness hallucination* (Zhou et al., 2023; Huang et al.; Es et al., 2024). It has emerged as a critical challenge for current LLMs, especially in scenarios where their parametric knowledge is insufficient or outdated.

A growing body of work has emerged to address faithfulness hallucination. Current approaches can be broadly categorized as follows: *(1) post-training-based methods* (Song et al., 2025; 2024; Bi et al., 2025; Liu et al., 2025) employ supervised fine-tuning and direct preference optimization (Rafailov et al., 2023) to enhance faithfulness. However, these methods often necessitate costly human or stronger LLM supervision (e.g., GPT-4). Meticulously creating thousands to tens of thousands of training examples incurs significant annotation costs. *(2) decoding strategy-based methods* (Gema et al., 2024; Shi et al., 2024) alleviate faithfulness hallucinations through a plug-and-play approach that can be easily adapted to newly developed LLMs. However, they typically double the inference computation by requiring parallel processing with perturbed and natural inputs.

To overcome these limitations, we propose Self-Supervised Faithfulness Optimization (SSFO). SSFO offers two advantages: (1) a self-supervised faithfulness alignment framework with a minor post-training cost (hundreds of self-generated examples), which helps SSFO adapt easily to newly developed LLMs. (2) no additional inference burden, which is crucial for lightweight deployment on edge devices (Yu et al., 2024). To generate the training signal for faithfulness, we leverage the model's own differential behavior when its knowledge access is altered. As shown in Fig. 1, we

leverage the model itself to generate pairs of preference data: the preferred response is generated from the query with retrieved context, while the dispreferred response is generated from the query alone, relying solely on the model's parametric knowledge. We then apply DPO (Rafailov et al., 2023) training to align the model toward enhanced faithfulness. Our results show that SSFO attains contextual faithfulness comparable to both *post-training-based methods* (Song et al., 2025; Bi et al., 2025; Liu et al., 2025) and *decoding strategy-based methods* (Gema et al., 2024).

To understand the underlying mechanism of self-supervised faithfulness alignment, we show that it can be attributed to the likelihood displacement phenomenon (Razin et al., 2025). We provide both a gradient-based analysis and empirical results demonstrating that likelihood displacement transfers probability mass from parametric-based tokens to context-aligned tokens, making the alignment well-grounded. Building on this insight, we adapt the DPO loss with a weighting scheme (SSFO-$\lambda$) to enhance this beneficial displacement and strengthen faithfulness alignment.

Results show that SSFO-$\lambda$ achieves state-of-the-art faithfulness, improving performance by an average of 12% on LLaMA-3 and 27% on Mistral across faithfulness metrics relative to the instruct baseline. SSFO and SSFO-$\lambda$ also deliver superior generalization, improving cross-lingual contextual faithfulness across diverse LLMs. Moreover, since trained on only hundreds of self-generated examples, they preserve LLM's general instruction following ability and avoid the catastrophic forgetting common in more extensive fine-tuning (Kirkpatrick et al., 2017; Dong et al., 2023; 2024).

Overall, our contributions can be summarized as follows:

- We introduce SSFO, a self-supervised method for LLM faithfulness alignment. SSFO leverages self-generated data during training, requiring no human annotations, superior LLM models, or ground-truth labels; the training signal derives entirely from contrasting the model's own parametric knowledge (as dispreferred examples) against retrieved knowledge (as preferred examples). We show that faithfulness alignment can be achieved via self-supervision.

- We analyze the alignment process through the lens of likelihood displacement and provide empirical evidence that probability mass shifts from parametric to context-grounded tokens. Motivated by this finding, we investigate an easy-to-implement variant (SSFO-$\lambda$) that explicitly encourages likelihood displacement and further boosts faithfulness alignment.

- We conduct comprehensive evaluations across diverse LLMs and benchmarks. Results show that SSFO achieves state-of-the-art faithfulness and superior generalization, including robust cross-lingual contextual faithfulness. Moreover, since trained with only hundreds of self-generated examples, SSFO preserves LLM's general instruction-following ability, avoiding the catastrophic forgetting common in more extensive fine-tuning.

## 2 PRELIMINARIES

**Direct Preference Optimization (DPO)** (Rafailov et al., 2023): RLHF is computationally expensive (Cheng et al., 2023; Yuan et al., 2023) and can suffer from instabilities (Song et al., 2023; Go et al., 2023). DPO bypasses both explicit reward estimation and performing reinforcement learning to learn the policy using a single maximum likelihood objective. The DPO loss is defined as:

$$\mathcal{L}_{\text{DPO}}(\pi_\theta; \pi_{\text{ref}}) = -\mathbb{E}_{(x,y_w,y_l)\sim\mathcal{D}}\left[\log\sigma\left(\beta\log\frac{\pi_\theta(y_w\mid x)}{\pi_{\text{ref}}(y_w\mid x)} - \beta\log\frac{\pi_\theta(y_l\mid x)}{\pi_{\text{ref}}(y_l\mid x)}\right)\right], \quad (1)$$

where $(x, y_w, y_l)$ represents a data sample from dataset $\mathcal{D}$ consisting of a prompt $x$, a preferred completion $y_w$, and an dispreferred completion $y_l$. $\pi_\theta$ is the policy model undergoing optimization, and reference model $\pi_{\text{ref}}$ is the original state of the model before optimization. The hyperparameter $\beta$ controls the difference between policy model $\pi_\theta$ and reference model $\pi_{\text{ref}}$.

**Likelihood Displacement** (Razin et al., 2025; Pal et al., 2024; Tajwar et al., 2024): Likelihood displacement is a counterintuitive phenomenon observed during direct preference optimization, where the probabilities for the preferred response $\pi_\theta(y_w\mid x)$ and the dispreferred response $\pi_\theta(y_l\mid x)$ both decrease, while the margin between them widens. Since $y_w$ is typically the (almost) optimal response (e.g., human-written or from a superior model), this reduction is problematic. Recent work (Yang et al., 2025; Gupta et al., 2025) aims to alleviate this phenomenon.

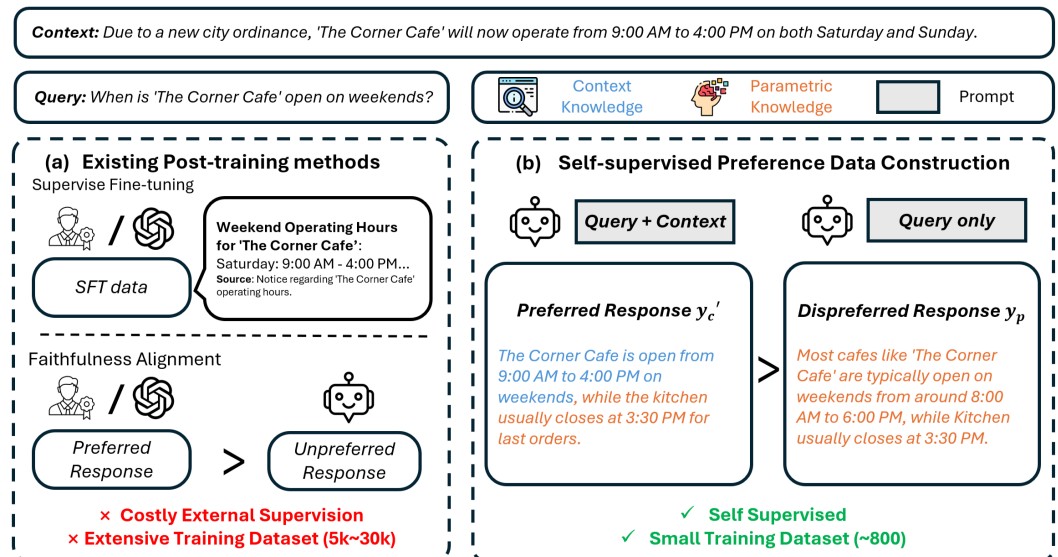

Figure 1: **(a)**: Existing post-training methods rely on human annotators or superior LLM models to construct SFT or preference datasets, resulting in heavy labeling costs and lengthy post-training processes. **(b)** SSFO leverages the model itself to generate preference data: Given query $x$, it generates a context-grounded response $y'_c$ (with external knowledge) and a parametric-based response $y_p$ (query only). SSFO reduces faithfulness hallucination without external supervision and incurs negligible post-training costs.

## 3 METHODOLOGY

In this section, we describe the methodology of the proposed Self-Supervised Faithfulness Optimization (SSFO). SSFO leverages self-supervised data construction and preference alignment training to reduce faithfulness hallucination in language models. Our goal is to train models to prioritize faithfulness to the provided external context over their internal parametric knowledge. This prioritization is critical for robust RAG systems.

### 3.1 SELF-SUPERVISED PREFERENCE DATA CONSTRUCTION

Existing approaches (Song et al., 2025; 2024; Bi et al., 2025) employ DPO to mitigate faithfulness hallucinations and rely on curated preference data, often from human annotators or superior LLM models like GPT-4, as shown in Fig. 1 (a). Although effective, these approaches incur substantial data annotation costs and post-training overhead.

To address this challenge, we propose a self-supervised data construction method that avoids external labeling or supervision, as shown in Fig. 1 (b). Our key idea is to exploit the LLM's own responses under different knowledge-access conditions to construct preference pairs. Specifically, we generate two types of outputs for preference optimization:

**Construction of preferred response:** We provide the model $\pi_\theta$ with the query $x$ and the retrieved context $c$ to construct preferred responses, i.e., $y \sim \pi_\theta(\cdot \mid x, c)$. Given the known faithfulness hallucination of LLMs (i.e., blend parametric knowledge and external context when generating responses) (Song et al., 2025; Niu et al., 2024; Bao et al., 2024), we denote this partially faithful response as $y'_c$.

**Construction of dispreferred response:** We provide the model with the query $x$ only, omitting the external context $c$. The model generates a response based solely on its parametric knowledge: $y \sim \pi_\theta(\cdot \mid x)$. We denote this response as $y_p$, which reflects the model's internal knowledge and is more susceptible to hallucinations due to the absence of grounding in retrieved information.

The preference data pairs $(y'_c, y_p)$ thus establish the context-grounded response as the positive example, and the parametric knowledge-based response as the negative example.

## 3.2 SELF-SUPERVISED FAITHFULNESS OPTIMIZATION

We perform DPO on the generated preference dataset $(y'_c, y_p)$ to achieve faithfulness alignment. Specifically, given a language model $\pi_\theta$, we minimize the following loss:

$$\mathcal{L}(\pi_\theta; \pi_{\text{ref}}) = -\mathbb{E}_{(x,c,y'_c,y_p)\sim\mathcal{D}} \left[ \log \sigma \left( \beta \log \frac{\pi_\theta(y'_c \mid x, c)}{\pi_{\text{ref}}(y'_c \mid x, c)} - \beta \log \frac{\pi_\theta(y_p \mid x, c)}{\pi_{\text{ref}}(y_p \mid x, c)} \right) \right]. \quad (2)$$

This objective encourages the model to increase the likelihood of the context-grounded response $y'_c$ while penalizing the parametric knowledge-based response $y_p$. The underlying principle is that $y'_c$, generated when conditioned on the external context, is generally more faithful than $y_p$, which relies solely on the model's internal parametric knowledge. By widening this preference margin, the model learns to prioritize contextual information over its internal knowledge, thereby mitigating faithfulness hallucinations without costly external supervision.

In practice, training on a few hundred instances yields significant improvements in faithfulness, outperforming methods that rely on human or superior LLM-generated training data (Section 4.3).

## 3.3 ANALYZING AND ENCOURAGING LIKELIHOOD DISPLACEMENT IN SELF-SUPERVISED FAITHFULNESS OPTIMIZATION

Empirical studies (Section 4.1) show that although $y'_c$ is an imperfect answer generated by $\pi_{\text{ref}}$, training with SSFO leads to a policy model $\pi_\theta^*$ that can significantly outperform $\pi_{\text{ref}}$. We attribute these gains to a benign form of likelihood displacement (Razin et al., 2025; Pal et al., 2024; Tajwar et al., 2024). Specifically, we demonstrate that in the context-based question-answering setting (i.e., RAG setting), SSFO shifts probability mass from tokens associated with parametric knowledge to those grounded in external contextual information. This effect suppresses the parametric component in both $y'_c$ and $y_p$, favoring tokens grounded in the external context.

As shown in Fig. 2 (left), we observe a likelihood displacement phenomenon during optimization: the optimized model $\pi_\theta^*$ satisfies $P_{\pi_\theta^*}(y'_c|x, c) < P_{\pi_\theta}(y'_c|x, c)$ and $P_{\pi_\theta^*}(y_p|x, c) < P_{\pi_\theta}(y_p|x, c)$, i.e., probability mass is driven away from both the composite response $y'_c$ and the parametric response $y_p$.

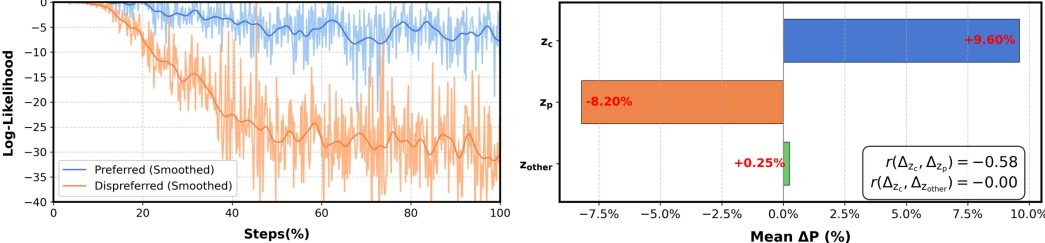

Figure 2: **Left**: Log-likelihood of preferred response $\pi_\theta(y'_c|x, c)$ versus dispreferred responses $\pi_\theta(y_p|x, c)$ over the course of SSFO optimization. **Right:** We compare the base instruct model and optimized model on MemoTrap (Liu & Liu, 2023) dataset and show the mean change for context-based tokens $z_c$ and parametric-based tokens $z_p$, revealing that optimization increases $\Delta P(z_c)$ while decreasing $\Delta P(z_p)$, $r$ denotes the Pearson correlation coefficient.

To understand where probability mass goes and ensure analytical tractability, we analyze the instantaneous update to the next-token distribution under gradient flow. Building on Theorem 5 of (Razin et al., 2025), the instantaneous change in the log-probability of an arbitrary token $z$ from vocabulary, conditioned on input context $(x, c)$, is given by:

$$\frac{d}{dt} \ln \pi_{\theta(t)}(z|x, c) \propto \langle W_z(t), W_{\text{token}(y'_c)}(t) - W_{\text{token}(y_p)}(t)\rangle, \quad (3)$$

Here, $W_z(t)$ denote the unembedding vector of token $z$ at training time $t$, while $W_{\text{token}(y'_c)}(t)$ is the unembedding vector of the token that the model is likely to generate given the context and $W_{\text{token}(y_p)}(t)$ corresponds to the token likely generated from the parametric-knowledge. In other words, **the larger the inner product** $\langle W_z(t), W_{\text{token}(y'_c)}(t) - W_{\text{token}(y_p)}(t)\rangle$, **the more positive the change in** $\pi_{\theta(t)}(z|x, c)$.

Let $V(t) = W_{\text{token}(y'_c)}(t) - W_{\text{token}(y_p)}(t)$ denote the direction vector. We analyze how the probabilities of different types of output tokens $z$ vary by examining the inner product $\langle W_z(t), V(t) \rangle$.

- **Faithful Token $z_c$ (derived from context $c$):** With the LLM's inherent ability to follow external context (Lewis et al., 2020; Zhou et al., 2023; Gao et al.), when generating the preferred response $y'_c$ conditioned on $c$, the model is highly likely to produce tokens consistent with the context. Thus, $W_{\text{token}(y'_c)}$ is expected to be well aligned with $W_{z_c}$. In contrast, since $y_p$ reflects ungrounded, parametric-based generation, $W_{z_c}$ is likely unaligned with $W_{\text{token}(y_p)}$. Therefore, the inner product $\langle W_{z_c}(t), V(t) \rangle$ is expected to be large.

- **Parametric Token $z_p$ (derived from internal knowledge, potentially hallucinated):** The token $z_p$ is likely aligned with $W_{\text{token}(y_p)}$, reflecting the model's internal parametric memory. However, its alignment with $W_{\text{token}(y'_c)}$ is expected to be weak or negative. Consequently, $\langle W_{z_p}(t), V(t) \rangle$ is expected to be small.

- **Irrelevant Token $z_{\text{other}}$ (unrelated to context $c$ or parametric response $y_p$):** $W_{z_{\text{other}}}$ is unlikely to exhibit strong alignment with either the context-dependent $W_{\text{token}(y'_c)}$ or the internal knowledge-based $W_{\text{token}(y_p)}$. As a result, $\langle W_{z_{\text{other}}}(t), V(t) \rangle$ is expected to be small.

Let $\Delta P(z)$ denote the increase in probability for token $z$ due to likelihood displacement, proportional to $\frac{d}{dt} \ln \pi_{\theta(t)}(z|x,c)$. Based on the analysis of the alignment above, we have:

$$\langle W_{z_c}(t), V(t) \rangle \gg \langle W_{z_p}(t), V(t) \rangle \quad \text{and} \quad \langle W_{z_c}(t), V(t) \rangle \gg \langle W_{z_{\text{other}}}(t), V(t) \rangle.$$

Therefore, in the Eq. (2) setting, the likelihood displacement mechanism preferentially transfers probability mass towards tokens $z_c$ that are consistent with the external context $c$. This constitutes a **benign likelihood displacement**, actively promoting faithfulness by reinforcing context-aligned generation while suppressing tokens derived from parametric knowledge or irrelevant content.

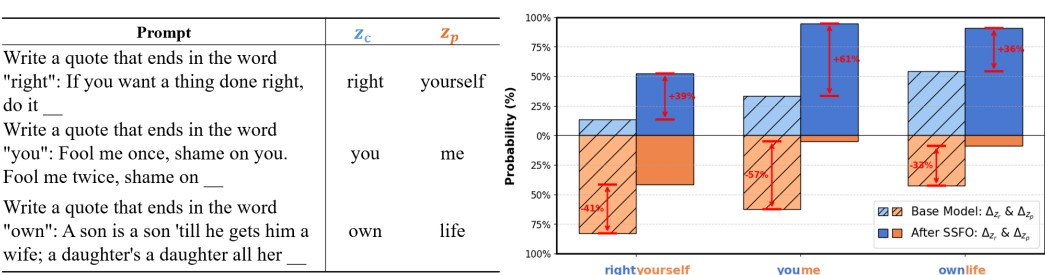

Figure 3: **Case study from the MemoTrap dataset illustrating benign likelihood displacement.** The probability mass shifts from the parametric knowledge based token $z_p$ to the external knowledge based token $z_c$ after SSFO optimization.

### 3.3.1 EMPIRICAL VALIDATION OF BENIGN LIKELIHOOD DISPLACEMENT

**Setting.** Our experiments utilize the MemoTrap dataset (Liu & Liu, 2023), designed to evaluate whether language models exhibit memorization traps. MemoTrap consists of instructions prompting the model to complete well-known proverbs with endings that deviate from the common completion. For instance, given the prompt "Write a quote that ends in the word 'right': If you want a thing done right, do it __", the instructed target completion is "right". In this context, the token "right" represents the external knowledge token $z_c$, while the commonly memorized completion token "yourself" is considered to be based on the parametric knowledge token $z_p$.

**The SSFO optimization induces a benign form of likelihood displacement.** As shown in Fig. 2 (Right), the probability of the faithful token $z_c$ increases after SSFO training. Furthermore, this rise is mirrored by a complementary fall in the probability of the parametric token $z_p$, producing a pronounced negative Pearson correlation ($r = -0.58$). Probabilities for all remaining vocabulary tokens remain essentially unchanged and show no discernible correlation with $z_c$. A case study illustrating this displacement is presented in Fig. 3.

### 3.3.2 ENCOURAGING BENIGN PROBABILITY DISPLACEMENT WITH SSFO-$\lambda$

As established in the previous analysis, the SSFO framework induces a **benign form of likelihood displacement**, in which probability mass shifts away from responses that rely on parametric knowledge to those grounded in the external context. To further promote this desirable effect, we introduce SSFO-$\lambda$, a variant that explicitly encourages this displacement through a single tuning parameter. The method is easy to implement, requiring only a rescaling of the DPO objective.

Prior approaches have mainly treated likelihood displacement as a **drawback** (Pal et al., 2024; Yang et al., 2025; Gupta et al., 2025; Xiao et al., 2024), since their "preferred" response $y_w$ is typically a high-quality, "golden" example (e.g., human-written), where reducing likelihood would indeed be harmful. However, our work explores using an imperfect, "silver" preferred response $y_c'$ generated by the reference model itself. As analyzed in Section 3.3, in this context-based question answering setting, encouraging the likelihood displacement proves to be an **advantage** for enhancing the model's faithfulness. Motivated by (Yang et al., 2025), we introduce a scaling factor $\lambda > 1$ to encourage the likelihood displacement during optimization:

$$\mathcal{L}_{\text{SSFO}-\lambda}\left(\pi_\theta; \pi_{\text{ref}}\right) = -\mathbb{E}_{(x,c,y_c',y_p)\sim\mathcal{D}}\left[\log\sigma\left(\beta\log\frac{\pi_\theta\left(y_c'\mid x,c\right)}{\pi_{\text{ref}}\left(y_c'\mid x,c\right)} - \lambda\cdot\beta\log\frac{\pi_\theta\left(y_p\mid x,c\right)}{\pi_{\text{ref}}\left(y_p\mid x,c\right)}\right)\right]. \tag{4}$$

**Empirical Validation of $\lambda$'s Effect.** As shown in Fig. 4, we investigate the impact of varying $\lambda$ from 1.0 to 1.5 across multiple context-based question-answering benchmarks: NQ-Swap, NQ-Open, MemoTrap, and ELI5. As $\lambda$ increases, we observe a consistent improvement in performance across all evaluated tasks. Pearson correlation coefficients $r$ reveal a positive relationship between $\lambda$ and performance on all datasets. For instance, span EM score on MemoTrap rises by 2.1 points (from 76.2% to 78.3%); NQ-Swap gains 1.2 points (from 81.2% to 82.5%). These results confirm that strategically amplifying the weight on the ungrounded (parametric) response via $\lambda > 1$ (to encourage benign likelihood displacement) indeed yields a more faithful response.

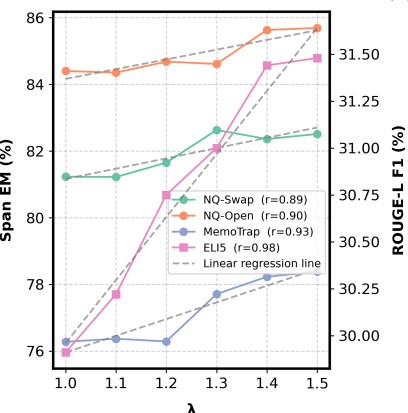

Figure 4: Correlation plot illustrating Span Exact Match scores for NQ-Swap, NQ-Open, and MemoTrap (scaled to the left y-axis) and ROUGE-L F1 scores for ELI5 (scaled to the right y-axis). Grey lines depict the regression trends. $r$ denotes Pearson correlation.

**Gradient Analysis.** To further understand how this modification encourages the desired displacement, we analyze the gradient of SSFO–$\lambda$ loss in Eq. (4). The gradient with respect to parameters $\theta$ is:

$$\nabla_\theta\mathcal{L}_{\text{SSFO}-\lambda} = -\mathbb{E}\Big[c_1'\big(\nabla_\theta\log\pi_\theta(y_c'|x,c) - \underbrace{\lambda\nabla_\theta\log\pi_\theta(y_p|x,c)}_{\text{decrease likelihood of } y_p}\big)\Big], \tag{5}$$

where $c_1'$ is a positive coefficient. We present a detailed derivation of Eq. (5) in Section C.2. Compared to the standard DPO update, Eq. (4) applies a stronger negative weight ($-\lambda$ where $\lambda > 1$) to the gradient component associated with the parametric response $y_p$. Therefore, this parameter leads to a more pronounced suppression of the likelihood of the parametric response during optimization.

## 4 EXPERIMENTS

**Datasets**: To comprehensively evaluate faithfulness, we assess model performance across several dimensions. (1) For evaluating **Robustness** against conflicting parametric knowledge, we follow prior work (Gema et al., 2024; Shi et al., 2024), using MemoTrap (Liu & Liu, 2023) and NQ-Swap (Longpre et al., 2021).(2) For **Response Quality**, we evaluate on the context-based short-form QA datasets NQ-Open (Lee et al., 2019) and SQuAD (Rajpurkar et al., 2016), as well as the long-form generation datasets ELI5 (Fan et al., 2019) and WikiPassageQA (Cohen et al., 2018). (3) To assess the generalization ability of the proposed methods, we benchmark **Cross-language Response Quality** using DuReader (He et al., 2018) and XQuAD (Artetxe et al., 2020), and **Instruction Following Ability** using FollowBench (Jiang et al., 2024).

**Metrics**: For short-form QA datasets (NQ-Open, NQ-Swap, MemoTrap, SQuAD), we adopt a standard zero-shot setting simulating a RAG scenario where the model answers queries based on the provided context. Performance is measured using the span Extraction Matching (span EM) score (a prediction is deemed correct if any segment of the generated output precisely matches one of the reference answers). For the long-form generation dataset ELI5, we report ROUGE scores (Lin, 2004) to quantify lexical overlap between the generated responses and the reference answers. We also report the LLM-Faithfulness Score (LFS). The LFS is calculated using GPT-4 (Achiam et al., 2023) to classify outputs as faithful, partially faithful, or unfaithful (see the prompt in Table 7), and the score is defined as the number of faithful generations divided by the total number of generations. For instruction following (FollowBench), we report Consistent Satisfaction Levels (CSL) (Jiang et al., 2024), which measures how many consecutive levels of instruction hardness a model can satisfy.

**Models and Baselines**: To ensure the generality of our approach, we conduct experiments using three families of open-source large language models: LLaMA 3 Instruct (Touvron et al., 2023), Qwen 2.5 Instruct (Yang et al., 2024), and Mistral Instruct. We compare the proposed method against the strong methods focused on improving faithfulness: CAD (Shi et al., 2024), DECORE (Gema et al., 2024), ChatQA (Liu et al., 2025), Trust-Align (Song et al., 2025), Context-DPO (Bi et al., 2025), and SCOPE (Duong et al., 2025). It is worth noting that SCOPE also claims to require no external supervision. However, SCOPE tunes task-specific models by using gold reference labels as positive examples and synthesizes unfaithful negatives from a mixture of a fine-tuned model and an unconditional pre-trained language model. This process needs to be repeated for each type of downstream task. In contrast, SSFO is entirely label-free and pursues a more direct alignment strategy, yielding broadly task-agnostic faithfulness gains.

## 4.1 FAITHFULNESS EVALUATION RESULTS

Table 1: **Faithfulness evaluation.** Comparison of SSFO and SSFO–$\lambda$ on **Robustness** under conflicting parametric knowledge (NQ-Swap, MemoTrap) and **Response Quality** on short-form (NQ-Open, SQuAD) and long-form (ELI5, WikiQA) datasets. Best results are shown in **bold**.

| Model | Method | Implement | Supervision | Robustness | | Response Quality | | | | | |
| | | | | NQ-Swap | Memo-Trap | NQ-Open | SQuAD | Eli5 | | WikiQA | |
| | | | | Span EM ↑ | Span EM ↑ | Span EM ↑ | Span EM ↑ | R-L F1 ↑ | LFS ↑ | R-L F1 ↑ | LFS ↑ |
| Llama-3-8B | **Instruct-Baseline** | \ | \ | 73.54% | 73.60% | 80.15% | 88.20% | 25.95% | 59.80% | 13.07% | 75.31% |
| | **Decoding-Strategy** | CAD | ✗ | 75.90% | 74.67% | 81.44% | 86.30% | 24.50% | 57.20% | 15.02% | 76.87% |
| | | DECORE | ✗ | 80.53% | 74.40% | 82.03% | 84.90% | 27.87% | 68.90% | 14.57% | 78.41% |
| | **Post-Training** | ChatQA | ✓ | 67.70% | 30.60% | 76.80% | 88.50% | 27.13% | 69.70% | 13.83% | 56.79% |
| | | Trust-Align | ✓ | 75.56% | 70.95% | 77.38% | 50.90% | 10.08% | 55.10% | 12.99% | 76.19% |
| | | Context-DPO | ✓ | 82.76% | 72.90% | 82.86% | 89.90% | 27.19% | 66.40% | 11.00% | 76.13% |
| | | SCOPE | ✗ | 76.72% | 74.26% | 80.38% | 68.80% | 22.41% | 60.20% | 15.69% | 76.46% |
| | | **SSFO** | ✗ | 81.23% | 76.28% | 84.40% | 89.00% | 29.91% | 71.40% | 13.98% | 75.72% |
| | | **SSFO-$\lambda$** | ✗ | **82.81%** | **78.38%** | **85.69%** | **90.90%** | **31.48%** | **72.30%** | **15.53%** | **79.01%** |
| Qwen2.5-7B | **Instruct-Baseline** | \ | \ | 79.35% | 54.19% | 82.29% | 90.30% | 23.11% | 41.30% | 15.33% | 68.72% |
| | **Decoding-Strategy** | CAD | ✗ | 79.78% | 63.10% | 84.29% | 85.90% | 18.10% | 49.80% | 14.28% | 26.75% |
| | | DECORE | ✗ | 81.93% | 54.56% | 83.76% | 82.80% | 26.83% | 53.60% | 14.49% | 71.66% |
| | **Post-Training** | Trust-Align | ✓ | 79.69% | 53.71% | 77.93% | 80.30% | 15.67% | 50.70% | **16.30%** | 73.84% |
| | | Context-DPO | ✓ | 82.13% | 55.34% | 83.13% | 91.80% | 23.81% | 49.30% | 15.23% | 72.84% |
| | | SCOPE | ✗ | 79.75% | 44.90% | **87.98%** | 78.50% | **36.26%** | 60.20% | 16.18% | 51.03% |
| | | **SSFO** | ✗ | 84.18% | 57.66% | 83.88% | 92.00% | 24.49% | 54.60% | 15.96% | **79.01%** |
| | | **SSFO-$\lambda$** | ✗ | **84.88%** | **60.77%** | 84.48% | **93.30%** | 23.96% | **62.80%** | 15.60% | 74.07% |
| Mistral-7B | **Instruct-Baseline** | \ | \ | 67.76% | 34.34% | 79.13% | 84.80% | 23.38% | 52.10% | 18.34% | 59.67% |
| | **Decoding-Strategy** | CAD | ✗ | 75.26% | 22.57% | 80.75% | 89.00% | 24.57% | 48.30% | 18.19% | 32.92% |
| | | DECORE | ✗ | 78.17% | 30.68% | 86.52% | 85.30% | 25.24% | 62.50% | **22.07%** | 67.13% |
| | **Post-Training** | Context-DPO | ✓ | 79.62% | 33.20% | 80.68% | 86.50% | 24.55% | 66.30% | 15.29% | 63.20% |
| | | SCOPE | ✗ | 49.58% | 15.87% | 64.71% | 54.00% | 27.06% | 63.40% | 15.47% | 60.29% |
| | | **SSFO** | ✗ | **86.66%** | 37.22% | 87.53% | **89.00%** | 30.43% | 80.60% | 16.67% | 63.79% |
| | | **SSFO-$\lambda$** | ✗ | 85.48% | **46.91%** | **90.32%** | 88.50% | **33.58%** | **88.10%** | 19.64% | **69.96%** |

*We present the results for varying model sizes in Table 8.

**SSFO and SSFO-$\lambda$ deliver strong faithfulness across multiple datasets and models**, as shown in Table 1: Both variants of Self-Supervised Direct Preference Optimization (SSFO and SSFO-$\lambda$) substantially improve contextual faithfulness over the instruct baseline. For example, SSFO markedly improves **Robustness**, raising NQ-Swap Span EM scores by 18.9% for Mistral, and boosts **Response Quality** on NQ-Open for Llama from 80.15% to 84.40%. SSFO-$\lambda$ further increases scores on Llama and achieves top scores across all metrics for this model. Overall, SSFO-$\lambda$ attains state-of-the-art performance compared to both post-training-based baselines and decoding strategy-based baselines. These results indicate that explicitly encouraging benign likelihood displacement not only enhances faithfulness but also improves overall response quality.

Table 2: **A case study on faithfulness improvement.** Comparing the instruct baseline, Context-DPO, and SSFO on equivalent English–Spanish (ES) query–answer pairs. This example from XQuAD (Artetxe et al., 2020) demonstrates that SSFO improves faithfulness over the instruct baseline and also preserves faithfulness cross-lingually. **Blue** texts highlight context-grounded information; **Red** texts mark unfaithful content.

| | |
|---|---|
| *Query* | **(EN)** What is Cultural Imperialism often referred to as? |
| | **(ES)** *¿A qué se refiere a menudo el imperialismo cultural?* |
| *Context* | **(EN)** ...**Cultural imperialism is when a country's influence is felt in social and cultural circles, i.e. its soft power**, such that it changes the moral, cultural and societal worldview of another... |
| | **(ES)** ...***El imperialismo cultural es cuando la influencia de un país se siente en los círculos sociales y culturales, es decir, su poder blando***, *de tal manera que cambia la cosmovisión moral, cultural y social de otro...* |
| *Instruct Model* | **(EN)** Cultural Imperialism is often referred to as cultural hegemony or cultural dominance.    **Hallucination** |
| *Context-DPO* | **(EN)** Cultural Imperialism is often referred to as the spread of a dominant culture's influence. This is also known as the use of soft power.    **Partially Context-Faithful** |
| | **(ES)** *El imperialismo cultural suele aludir a la influencia de la cultura de un país sobre la esfera sociocultural de otro, transformando su cosmovisión.*    **Hallucination** |
| *SSFO* | **(EN)** A country's influence is felt in social and cultural circles, i.e. its soft power, such that it changes the moral, cultural, and societal worldview of another...    **Context-Faithful** |
| | **(ES)** *La influencia de un país se siente en los círculos sociales y culturales, es decir, su poder blando, de tal manera que cambia la cosmovisión moral, cultural y social de otro...*    **Context-Faithful** |

Table 3: **Cross-language faithfulness and instruction-following evaluation.** Comparison of SSFO and SSFO–$\lambda$ on cross-language context-based QA benchmarks (XQuAD—Spanish, DuReader—Chinese) and instruction-following (FollowBench).

| Model | Method | Implement | Training Data Required | Cross-language Response Quality | | Instruction Following |
|---|---|---|---|---|---|---|
| | | | | **XQuAD(ES)** | **DuReader(CN)** | **FollowBench** |
| | | | | Span EM ↑ | Span EM ↑ | CSL ↑ |
| **Llama-3-8B** | **Instruct-Baseline** | \ | \ | 78.60% | 78.80% | 2.54 |
| | **Decoding-Stratagy** | CAD | \ | 70.34% | 76.57% | 0.92 |
| | | DECORE | \ | 81.87% | 79.89% | 2.46 |
| | **Post-Training** | ChatQA | ∼30k | 77.98% | 72.05% | 1.04 |
| | | Trust-Align | ∼15k | 20.17% | 8.12% | 0.12 |
| | | Context-DPO | ∼5k | 83.03% | 84.40% | 2.46 |
| | | SCOPE | ∼5k | 69.70% | 73.90% | 0.16 |
| | | **SSFO** | **∼800** | 83.10% | **84.90%** | **2.70** |
| | | **SSFO-$\lambda$** | **∼800** | **84.12%** | 83.56% | 2.50 |
| **Qwen2.5-7B** | **Instruct-Baseline** | \ | \ | 78.90% | 81.50% | 2.68 |
| | **Decoding-Stratagy** | CAD | \ | 71.85% | 76.71% | 1.22 |
| | | DECORE | \ | 80.08% | 76.57% | 2.56 |
| | **Post-Training** | Trust-Align | ∼15k | 75.21% | 73.61% | 0.58 |
| | | Context-DPO | ∼5k | 79.92% | 82.78% | 2.64 |
| | | SCOPE | ∼5k | 84.96% | 89.27% | 0.42 |
| | | **SSFO** | **∼800** | 79.83% | 83.27% | **2.70** |
| | | **SSFO-$\lambda$** | **∼800** | **81.76%** | **87.72%** | 2.62 |

## 4.2 GENERALIZATION ACROSS TASKS AND LANGUAGES

**SSFO enhances multi-language faithfulness.** We evaluate the generalization ability of SSFO in Table 3, and results show it can improve cross-lingual faithfulness using only an English-based training set. For instance, on Llama, SSFO increases Span EM scores by 6.10% on DuReader (Chinese) and 5.52% on XQuAD (Spanish) compared to the instruct baseline. In contrast, heavily supervised methods like ChatQA (Liu et al., 2025) and Trust-Align (Song et al., 2025) exhibit decreased performance on these non-English QA datasets. **This shows that by training the model to prioritize context knowledge over parametric knowledge, it learns a principle of contextual adherence that can transfer across languages.**

**SSFO minimally impacts instruction following capability.** Requiring only a few hundred self-supervised data examples, SSFO largely preserves, and even slightly enhances, the model's instruction following capabilities. The CSL scores on FollowBench indicate that models fine-tuned with SSFO

retain comparable general instruction-following ability to the original base instruction models. We provide several cases that SSFO retains strong general instruction following ability under context-based scenarios in Table 4. In contrast, other post-training approaches, such as Trust-Align (Song et al., 2025), improve faithfulness at the cost of degrading general generative abilities (e.g., CSL score decreases from 2.54 to 0.12 on LLaMA-3-8B-Instruct).

Table 4: **Case study from FollowBench (Jiang et al., 2024)**. SSFO retains strong instruction-following capabilities under a context-based, composite NLP task. In contrast, the heavily post-trained Trust-Align model (Song et al., 2025) fails, demonstrating a loss of general capabilities.

| | |
|---|---|
| ***Query*** | Analyze the provided text to pinpoint a category and the sentiment (positive, neutral, or negative) it emanates. Additionally, perform named entity recognition to emphasize notable entities and also identify the core topic discussed. Select the category from this array: company, educational institution, artist, athlete, office holder, means of transportation, building, natural place, village, animal, plant, album, film, or written work. |
| ***Context*** | Michael DenDekker - Michael G. DenDekker (born July 11, 1961) is an assemblyman for the state of New York's 34th district which includes the neighborhoods of Woodside, Jackson Heights, and East Elmhurst, all in the borough/county of Queens. |
| ***Trust Align*** | I apologize, but I couldn't find an answer to your question in the search results. **General Ability Lost** |
| ***SSFO*** | **Category:** Office Holder; **Sentiment:** Neutral ; **Notable entities:** Michael DenDekker, State of New York, 34th district, Woodside... **Core topic discussed:** Michael DenDekker's role as an assemblyman for the state of New York's 34th district. **Rationale:** Note that the text does not contain any information about the office holder's achievements, opinions, or personal life, which is why the sentiment is neutral... **General Ability Retained** |

## 4.3 DATA EFFICIENCY ANALYSIS

To measure how many self-supervised preference examples SSFO actually needs, we subsample the training dataset in 10% increments. As shown in Fig. 5, we evaluate the average performance gain (an average improvement over the base instructed model). SSFO models cross the 85% performance threshold by approximately 50–60 % of the data (400–500 examples). We attribute this efficiency stems from using self-generated data, which avoids the stylistic distribution mismatch often caused by external data from human annotators or superior LLM models. Since the training data inherently matches the model's native response style, optimization can focus on improving faithfulness. We compare the training examples from SSFO with other post-training methods in Table 6.

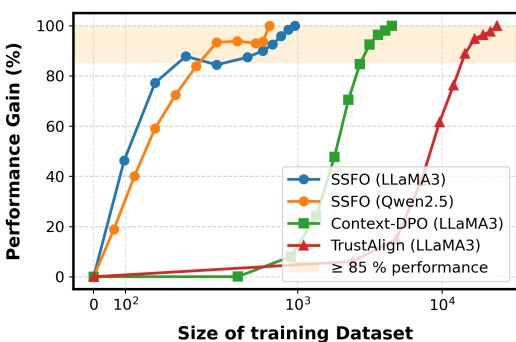

Figure 5: Data efficiency study: SSFO requires about 60% of data (400–500 examples) to achieve 85% of the total performance gain over the instruct baseline.

## 5 CONCLUSION

This work addressed the critical challenge of faithfulness hallucination in RAG systems, where existing methods often introduce significant computational overhead or rely on costly external supervision. We introduced SSFO, an efficient self-supervised alignment approach that leverages the model's own outputs to build preference pairs by comparing responses generated with retrieved context to responses based only on parametric knowledge. Our analysis shows the alignment proceeds through a benign form of likelihood displacement, which shifts probability mass from parametric-based tokens to context-aligned ones. Motivated by this finding, we proposed SSFO-$\lambda$, a variant that amplifies this beneficial displacement and further enhances faithfulness. Our experiments across diverse benchmarks show that SSFO and SSFO-$\lambda$ significantly enhance model faithfulness and robustness against parametric knowledge, achieving state-of-the-art performance compared to existing methods. Furthermore, SSFO exhibits strong generalization capabilities, improving faithfulness even in cross-lingual settings using only English training data, while preserving the model's general instruction-following abilities.

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

## A    APPENDIX

## B    RELATED WORK

### B.1    FAITHFULNESS HALLUCINATION OF LARGE LANGUAGE MODELS

Hallucination in LLMs can be generally categorized into two types: factuality hallucination, where generated content deviates from established world knowledge (e.g., claiming "Mars has oceans"), and faithfulness hallucination, where the generated response is inconsistent with the provided context(e.g., misrepresenting a source document's information). (Huang et al.)

Current methods to address faithfulness hallucination primarily fall into two categories:

- *Post-training-based methods* rely on supervised fine tuning (Touvron et al., 2023; Hu et al., 2022) or preference alignment (Rafailov et al., 2023). Liu et al. (2025) propose a two-stage instruction tuning method and create a dataset (including human annotation) that aims at enhancing LLM's capability of integrating external context. For alignment-based methods, one key factor lies in creating the preference dataset: Song et al. (2025) uses GPT-4 (Achiam et al., 2023) to generate a well cross-referenced response as the positive answer and uses Llama2 (Touvron et al., 2023) to generate negative response results in an alignment dataset of 15K samples. RAG-HAT (Song et al., 2024) prompts GPT-4 to correct hallucinations in the response, which uses as positive response and the original response as the negative one. Context-DPO creates preference data by perturbing a knowledge graph and employs GPT-4 to generate counterfactual context (Bi et al., 2025). While these methods can yield more customized responses, they often demand costly supervision from humans or advanced LLM models and can lead to extensive post-training processes that may cause catastrophic forgetting (Kirkpatrick et al., 2017; Lin et al., 2024), thereby undermining the model's generalization capabilities

- *Decoding strategy-based methods*: In (Shi et al., 2024), the author presents context-aware decoding (CAD), which follows a contrastive output distribution that amplifies the difference between the output probabilities when a model is used with and without context. DECORE (Gema et al., 2024) extends this framework to masking retrieval heads to induce faithfulness hallucinations, followed by a dynamic entropy-controlled contrastive decoding to penalize uncertain outputs. While these methods are training-free and adaptable, they often significantly increase the inference burden, typically by requiring parallel processing

To overcome these challenges, this paper introduces Self-Supervised Faithfulness Optimization (SSFO), a self-supervised alignment method that enhances faithfulness without introducing external supervision or additional inference burden. To our knowledge, the most closely related work is SCOPE (Duong et al., 2025), which tunes task-specific models by using gold reference labels as positive examples and synthesizes unfaithful negatives via a noisy, token-level mixture of a fine-tuned model and an unconditional pre-trained LM. This process must be repeated for each new downstream task, limiting its generalizability across diverse RAG scenarios. In contrast, SSFO is entirely label-free and pursues a more direct alignment strategy. It generates its own preference data by contrasting the model's output generated with retrieved context against the output generated without context. Moreover, we analyze this self-supervised alignment process, demonstrating that it leverages a benign form of likelihood displacement to enhance faithfulness. Overall, SSFO learns a broadly task-agnostic principle of contextual adherence, yielding significant faithfulness gains without requiring ground-truth labels or being rebuilt for each new task.

### B.2 Direct Preference Optimization and Likelihood Displacement

RLHF (Ouyang et al., 2022; Bai et al., 2022) requires fitting a reward model to a dataset of human (or AI) preferences, and then training the language model to maximize the reward, which is computationally expensive and can suffer from instabilities. This has led to the rise of direct preference optimization (DPO) (Rafailov et al., 2023) . DPO implicitly optimizes the same objective as RLHF algorithms but is easy to implement and straightforward to train.

Likelihood displacement (Razin et al., 2025) refers to the counterintuitive phenomenon where, during direct preference alignment, while the gap between preferred responses and dispreferred responses increases, they both decrease. Such a phenomenon is unwanted since the preferred response is derived from a human annotator or a strong AI model. To alleviate this problem, DPOP (Pal et al., 2024) design a modified DPO loss function to penalizes reducing the probability of the positive completion; AlphaPO (Gupta et al., 2025) introduce a parameter to adjust the shape of the reward function beyond standard log rewards, providing fine control over the likelihood displacement; DPO-Shift (Yang et al., 2025) adds a real-valued function to controllably shift the distribution of the preferred probability.

Existing approaches typically assume the preferred response is a "golden" label and aim to alleviate likelihood displacement. In contrast, SSFO optimizes the model using self-supervised preference data, which can be considered "silver" labels, yet still provides a clear supervisory signal towards faithfulness. This work demonstrates that, in the RAG setting, likelihood displacement can be a benign phenomenon and can even be encouraged to benefit the faithfulness alignment process.

## C Mathematical Derivations

### C.1 Likelihood displacement analysis for $\mathcal{L}_{\text{SSFO}-\lambda}$

Let $\sigma$ denote the logistic function. Define the chosen-likelihood target and the (smoothed) reward-margin target:

$$\omega_1(\theta) = \mathbb{E}[\log \pi_\theta(y'_c \mid x, c)], \qquad \omega_2(\theta) = \mathbb{E}\Big[\sigma\Big(\gamma\log\frac{\pi_\theta(y'_c|x,c)}{\pi_{\text{ref}}(y'_c|x,c)} - \gamma\log\frac{\pi_\theta(y_p|x,c)}{\pi_{\text{ref}}(y_p|x,c)}\Big)\Big].$$

Consider the SSFO-$\lambda$ loss:

$$\mathcal{L}_{\text{SSFO-}\lambda}(\theta) = -\mathbb{E}\Big[\log\sigma\Big(\beta\log\frac{\pi_\theta(y'_c|x,c)}{\pi_{\text{ref}}(y'_c|x,c)} - \lambda\beta\log\frac{\pi_\theta(y_p|x,c)}{\pi_{\text{ref}}(y_p|x,c)}\Big)\Big],$$

Let $\theta_{t+1} = \theta_t - \eta\,\nabla\mathcal{L}(\theta_t)$, and define the one-step gaps between SSFO-$\lambda$ and vanilla SSFO (i.e., $\lambda = 1$):

$$g_i(t+1) = \omega_i(\theta_{t+1})\big|_{\text{SSFO-}\lambda} - \omega_i(\theta_{t+1})\big|_{\text{SSFO}}, \quad i \in \{1,2\}.$$

Following Theorem 2.1 of (Yang et al., 2025), for a single gradient step and to first order,

$$g_1(t+1) = (1-\lambda)\,u_1, \qquad g_2(t+1) = (1-\lambda)\,u_2,$$

where $u_1 > 0$ and $u_2 < 0$.

If $\lambda > 1$, then $1 - \lambda < 0$, hence

$$g_1(t+1) < 0 \quad \text{and} \quad g_2(t+1) > 0.$$

Thus, the chosen likelihood $\omega_1$ decreases while the margin $\omega_2$ increases, i.e., choosing $\lambda > 1$ encourages likelihood displacement.

## C.2 GRADIENT DERIVATION FOR $\mathcal{L}_{\text{SSFO}-\lambda}$

The loss function for SSFO-$\lambda$ is given by:

$$\mathcal{L}_{\text{SSFO}-\lambda}(\pi_\theta, \pi_{ref}) = -\mathbb{E}_{(x,c,y'_c,y_p)\sim\mathcal{D}_{pref}}\left[\log\sigma(u)\right]$$

where

$$u := \beta\log\frac{\pi_\theta(y'_c|x,c)}{\pi_{ref}(y'_c|x,c)} - \lambda\cdot\beta\log\frac{\pi_\theta(y_p|x,c)}{\pi_{ref}(y_p|x,c)}$$

The gradient with respect to $\theta$ is:

$$\nabla_\theta\mathcal{L}_{\text{SSFO}-\lambda} = -\mathbb{E}\left[\frac{\sigma'(u)}{\sigma(u)}\nabla_\theta u\right]$$

Using the properties of the sigmoid function $\sigma'(x) = \sigma(x)(1-\sigma(x))$ and substituting $-u$, the gradient simplifies to:

$$\nabla_\theta\mathcal{L}_{\text{SSFO}-\lambda} = -\mathbb{E}\left[\sigma\left(\lambda\cdot\beta\log\frac{\pi_\theta(y_p|x,c)}{\pi_{ref}(y_p|x,c)} - \beta\log\frac{\pi_\theta(y'_c|x,c)}{\pi_{ref}(y'_c|x,c)}\right)\right.$$
$$\left.\times\left(\beta\nabla_\theta\log\pi_\theta(y'_c|x,c) - \lambda\cdot\beta\nabla_\theta\log\pi_\theta(y_p|x,c)\right)\right] \tag{6}$$

Let $c'_1$ be defined as:

$$c'_1 := \beta\sigma\left(\lambda\cdot\beta\log\frac{\pi_\theta(y_p|x,c)}{\pi_{ref}(y_p|x,c)} - \beta\log\frac{\pi_\theta(y'_c|x,c)}{\pi_{ref}(y'_c|x,c)}\right)$$

Then the final gradient form is:

$$\nabla_\theta\mathcal{L}_{\text{SSFO}-\lambda} = -\mathbb{E}\left[c'_1\left(\nabla_\theta\log\pi_\theta(y'_c|x,c) - \lambda\nabla_\theta\log\pi_\theta(y_p|x,c)\right)\right]$$

This matches the target formula.

## D  IMPLEMENTATION DETAILS

### D.1  DATASETS

We utilize a variety of datasets to comprehensively evaluate the proposed SSFO method across different aspects of faithfulness, response quality, and generalization to ensure consistently strong performance in a retrieval-augmented generation (RAG) setting.

- **MemoTrap** (Liu & Liu, 2023) is designed to reveal "memorisation traps" by pitting a well-known proverb against a context-correct but counter-habitual ending. *Example (prompt)*: "*Write a quote that ends in the word 'right': If you want a thing done right, do it ___*" – the context expects the completion *right*, not the cached continuation *yourself*.
- **NQ-Open** (Lee et al., 2019) is an open-domain QA benchmark provide with supporting passages. *Example*: *Passage: Vatican City.... is the smallest country in Europe by both area and population; Question: Which country has the smallest population in Europe?*" → *Vatican City*.
- **NQ-Swap** (Longpre et al., 2021) extends Natural-Questions with entity swaps to create conflicts between retrieved context and parametric memory. *Example*: Context states "*Ferraro is known for her portrayal of Grace Bowman in The Secret Life of the American Teenager*"; the query asks "*Who plays Grace in . . . ?*" – the correct answer is *Ferraro*, although parametric knowledge often yields *Molly Ringwald*.

- **SQuAD v1.1** (Rajpurkar et al., 2016) provides short passages with span-based questions. *Example*: *Passage: "Google was founded in 1998 by Larry Page and Sergey Brin"*; *"Question: Who founded Google?"* → *Larry Page; Sergey Brin.*
- **ELI5** (Fan et al., 2019) contains long-form, lay-audience explanations. *Example*: *"Why is the sky blue?"* expects a multi-sentence answer discussing Rayleigh scattering.
- **WikiPassageQA** (Cohen et al., 2018) is a collection designed for long-form, non-factoid answer passage retrieval. It contains thousands of questions with annotated answers. *Example*: *"What does s.h.i.e.l.d stand for?"* → *"The acronym originally stood for Supreme Headquarters, International Espionage, Law-Enforcement Division. "*
- **DuReader** (Chinese) (He et al., 2018) evaluates cross-lingual comprehension with Web passages. *Example (in English for illustration)*: *"Who wrote Dream of the Red Chamber?"* → *Cao Xueqin.*
- **XQuAD** (Spanish split) (Artetxe et al., 2020) probes zero-shot transfer to non-English languages. *Example (in English for illustration)*: *"Who was the first person to transmit radio waves across the Atlantic?"* → *Guglielmo Marconi.*
- **FollowBench** (Jiang et al., 2024) measures fine-grained instruction following. *Example*: *To enhance your time management skills, can you devise a method incorporating a mind map and featuring a touch of alliteration in the suggestion, ensuring each sentence contains no more than 15 words?*

## D.2 BASELINES

**Instruct Model** (Touvron et al., 2023; Yang et al., 2024): A vanilla instruction-tuned LLM queried with a standard retrieval-augmented generation (RAG) prompt.

**CAD** (Shi et al., 2024): A training-free decoding strategy that contrastive output distribution that amplifies the difference between the output probabilities when a model is used with and without context.

**DECORE** (Gema et al., 2024): A training-free decoding strategy that reduces hallucinations by contrasting outputs of the base LLM and a masked variant (retrieval heads suppressed) guided by conditional entropy. For comparison, we reproduce DECORE with the authors' open-source implementation.

**Trust-Align** (Shi et al., 2024): Builds GPT-4 "gold" answers cross-referenced to the retrieved context as positive samples and Llama outputs as negative samples, then performs DPO to steer the model toward faithful responses. For comparison, we use the official open-source model.

**ChatQA** (Liu et al., 2025): Enhances RAG and conversational QA via a two-stage instruction tuning method and a dense retriever optimized for dialogue, reducing deployment costs while matching query rewriting models. For comparison, we use the official open-source model.

**Context-DPO** (Bi et al., 2025): Improves context faithfulness by applying Direct Preference Optimization on the CONFIQA benchmark, which injects knowledge conflicts to mimic real RAG scenarios. For comparison, we use the official open-source model.

**SCOPE** (Duong et al., 2025): Tunes task-specific models using gold labels as positives while synthesizing negatives from a mixture of a fine-tuned and a pre-trained language model. Since SCOPE involves selecting specific training datasets for different tasks, we used the same dataset as our own method for a fair comparison in our experiments.

## D.3 DETAILS ON SELF-SUPERVISED PREFERENCE DATA CONSTRUCTION

Starting from the MS MARCO (Bajaj et al., 2018) corpus, we randomly sample and construct 900 (*query*, *context*) pairs, ensuring topic diversity and broad open-domain coverage. Using the prompts in Table 5, we construct both preferred and dispreferred responses from the base-instruct model, discarding empty outputs (e.g., responses such as "I do not know"). This yields a self-supervised preference dataset that reflects the model's native response style. In practice, training the base-instruct model on approximately 500 pairs already achieves over 85% of the final performance gain.

We compare generated preference data in Table 6, Trust-Align induces a large style gap—the preferred response is verbose and citation-driven, while the dispreferred one is concise and citation-light; Context-DPO performs closed-form QA with simple entity swaps (e.g., "microfiber cloth" vs.

Table 5: Prompts for self-supervised preference data generation and inference.

| **Prompt for preferred response** |
| --- |
| Based on the following context: |
| Context: **{Context}** |
| Question: **{Question}** |
| If you are not sure of the answer, please reply "I don't know". |
| **Prompt for dispreferred response** |
| Question: **{Question}** |
| If you are not sure of the answer, please reply "I don't know". |
| **Inference Prompt** |
| Based on the following context: **{Context}** |
| Answer the following question: **{Question}** |

"paper towel"); **SSFO (Open-Domain QA) preserves the model's native style for both preferred and dispreferred answers, enabling optimization to focus purely on faithfulness improvements without introducing a confounding style gap.**

Table 6: Comparison of responses generated by different post-training-based methods.

| Strategy | Preferred Response | Dispreferred Response |
| --- | --- | --- |
| **Trust-Align** | To safely clean your computer screen, first power off the device, as this makes dust and smudges easier to see and reduces the risk of damage [2][3]. Use only a clean, dry microfiber cloth as microfiber is specifically designed to clean screens gently without scratching them [1][3]. | First, turn off your laptop to prevent any damage to the screen. According to a guide by[1]How to Safely Clean Your Screen, this is essential for effective cleaning. .... **References: [1] How to Safely Clean Your Screen** |
| **Context-DPO** | Lightly wipe the screen with a clean, dry **microfiber cloth**. | Lightly wipe the screen with a clean, dry **paper towel**. |
| **SSFO** | Based on the passages, to safely clean your computer screen: 1. Wipe gently with a microfiber cloth, avoiding excessive pressure. 2. Do not use paper towels, kitchen towels, or household rags. | To safely clean your computer screen, you can use a soft, dry cloth or a microfiber cloth to gently wipe it. Avoid liquids, chemicals, or abrasive materials that may damage the surface. |

### D.4 TRAINING AND INFERENCE SETTINGS.

We fine-tune each model using a global batch size of 8 on 8×NVIDIA A6000 GPUs in BF16. Optimisation employs AdamW ($\beta_1$=0.9, $\beta_2$=0.95) with a peak learning rate of $1\times10^{-6}$, linearly warmed up from 0 over the first 10 steps and then held constant. For SSFO-$\lambda$, we set $\lambda = 1.5$ for Llama and Mistral, and $\lambda = 1.3$ for Qwen2.5. During both dataset generation and evaluation, we decode with a temperature of 0.7.

Table 7: Prompts for LLM-Faithfulness Score

| Prompt for using LLM-as-Judge to evaluate the faithfulness of long-form generation |
|---|
| **Task:**
You are a rigorous, no-nonsense evaluator of CONTEXT-faithfulness for retrieval-augmented generation. Read the CONTEXT and the QUESTION, then assess whether the MODEL_ANSWER is faithful to the CONTEXT when addressing the QUESTION.
**Definitions:**
- `"unfaithful"`: The answer contradicts the CONTEXT, or asserts key facts not supported by the CONTEXT (using outside/world knowledge, fabrications, or misquotes). If any contradiction exists, choose this.
- `"partially_faithful"`: The answer is mostly grounded but includes at least one unsupported or speculative claim, or mixes supported and unsupported details, or overgeneralizes beyond the CONTEXT without directly contradicting it.
- `"faithful"`: Every substantive claim used to answer the QUESTION is directly supported by the CONTEXT (allow paraphrases). Minor wording changes are fine; no extraneous unsupported facts.
**Inputs:**
- CONTEXT: `{CONTEXT}`
- QUESTION: `{QUESTION}`
- MODEL_ANSWER: `{MODEL_ANSWER}` |

# E  ADDITIONAL RESULTS

## E.1  EFFECT OF MODEL SCALE ON SSFO PERFORMANCE

Table 8: Impact of SSFO on robustness and response quality across varying LLM scales (1.5B–72B parameters).

| Instruct Model | Method | Robustness | | Response Quality | | | | |
|---|---|---|---|---|---|---|---|---|
| | | NQ-Swap | Memo-Trap | NQ-Open | SQuAD | Eli5 | | |
| | | Span EM↑ | Span EM↑ | Span EM↑ | Span EM↑ | R-1 F1↑ | R-2 F1↑ | R-L F1↑ |
| **Qwen2.5 1.5B** | Baseline | 65.78% | 25.44% | 79.17% | 88.80% | 22.11% | 4.26% | 19.40% |
| | SSFO | **79.65%** | **41.12%** | **83.88%** | **92.90%** | **28.10%** | **8.86%** | **24.87%** |
| **Qwen2.5 3B** | Baseline | 76.38% | 47.66% | 76.95% | 88.20% | 23.57% | 6.60% | 21.11% |
| | SSFO | **82.11%** | **59.12%** | **81.32%** | **92.40%** | **25.74%** | **9.16%** | **23.31%** |
| **Qwen2.5 7B** | Baseline | 79.35% | 54.19% | 82.29% | 90.30% | 23.08% | 6.06% | 20.55% |
| | SSFO | **84.18%** | **57.66%** | **83.88%** | **92.00%** | **24.63%** | **7.17%** | **21.83%** |
| **Qwen2.5 14B** | Baseline | 82.15% | 64.08% | 82.49% | 90.00% | 22.48% | 5.62% | 20.03% |
| | SSFO | **85.04%** | **66.52%** | **84.14%** | **92.80%** | **25.42%** | **8.19%** | **22.97%** |
| **Qwen2.5 72B** | Baseline | 81.84% | 66.99% | 83.50% | 91.20% | 21.85% | 5.26% | 19.45% |
| | SSFO | **87.51%** | **67.39%** | **84.48%** | **92.70%** | **22.46%** | **5.71%** | **19.96%** |

To assess the scalability of SSFO, we apply it to Qwen2.5 models with sizes from 1.5 B to 72 B parameters and report relative gains over each instruct baseline on five benchmarks (Table 8). Across all model sizes, SSFO consistently improves robustness—yielding +3 % to +14 % span EM gains—and enhances response quality—boosting closed-book QA and long-form generation metrics by up to +6 %. The largest relative improvements occur on smaller models (e.g., +13.9 % EM on Memo-Trap for Qwen2.5 1.5 B), while even the 72 B model sees steady gains (e.g., +5.6 % EM on NQ-Swap). These results demonstrate that SSFO delivers a stable, scalable enhancement to faithfulness and answer quality across a wide spectrum of LLM sizes.

## E.2  INSTRUCT FOLLOWING

In Table 9, we present detailed sub-metrics for several faithfulness-enhancement methods to evaluate their impact on LLMs' overall instruction-following capabilities. Among these approaches, SSFO not only best preserves general instruction adherence but also delivers the gains on Content-category tasks—a result we attribute to SSFO's ability to steer the model to attend more closely to the source text. **SSFO emerges as a practical technique for boosting faithfulness while maintaining the**

Table 9: Constraint Satisfaction Levels (CSL↑) on FollowBench across Llama-3-Instruct and Qwen2.5-7B-Instruct models. Results are broken down by Content, Situation, Style, Format, and Mixed constraint categories, as well as their overall mean.

| Model | Method | Implement | Supervision | Instruction Following (FollowBench) | | | | | |
|---|---|---|---|---|---|---|---|---|---|
| | | | | Content CSL ↑ | Situation CSL ↑ | Style CSL↑ | Format CSL ↑ | Mixed CSL ↑ | CSL Mean↑ |
| Llama-3-8B | Instruct-Baseline | \ | \ | 2.6 | 2.4 | 3.3 | 2.9 | 1.5 | 2.54 |
| | Decoding-Strategy | CAD | ✗ | 1.0 | 0.7 | 2.1 | 0.6 | 0.2 | 0.92 |
| | | DECORE | ✗ | 2.4 | 2.0 | 3.1 | 3.3 | 1.5 | 2.46 |
| | Post-Training | ChatQA | ✓ | 1.3 | 1.0 | 1.0 | 1.1 | 0.8 | 1.04 |
| | | Trust-Align | ✓ | 0.1 | 0.1 | 0.0 | 0.0 | 0.4 | 0.12 |
| | | Context-DPO | ✓ | 2.5 | 2.5 | 3.0 | 2.8 | 1.5 | 2.46 |
| | | SCOPE | ✗ | 0.4 | 0.2 | 0.1 | 0.1 | 0.0 | 0.16 |
| | | **SSFO** | ✗ | 2.8 | 2.7 | 3.2 | 3.2 | 1.6 | **2.70** |
| | | SSFO-$\lambda$ | ✗ | 2.3 | 2.4 | 3.1 | 3.1 | 1.6 | 2.50 |
| Qwen2.5-7B | Instruct-Baseline | \ | \ | 2.6 | 3.5 | 2.9 | 2.8 | 1.6 | 2.68 |
| | Decoding-Strategy | CAD | ✗ | 1.0 | 1.4 | 2.4 | 0.8 | 0.5 | 1.22 |
| | | DECORE | ✗ | 2.4 | 3.2 | 2.7 | 2.8 | 1.7 | 2.56 |
| | Post-Training | Trust-Align | ✓ | 0.5 | 1.5 | 0.3 | 0.1 | 0.5 | 0.58 |
| | | Context-DPO | ✓ | 2.6 | 3.3 | 3.0 | 2.9 | 1.4 | 2.64 |
| | | SCOPE | ✗ | 0.4 | 0.9 | 0.3 | 0.1 | 0.4 | 0.42 |
| | | **SSFO** | ✗ | 2.6 | 3.1 | 3.2 | 2.9 | 1.7 | **2.70** |
| | | SSFO-$\lambda$ | ✗ | 2.6 | 3.0 | 3.2 | 2.7 | 1.6 | 2.62 |

**model's effectiveness on general-purpose tasks.** For example, in a RAG-powered law assistant, in a legal research setting, SSFO can preserve verbatim case-law citations yet allows users to reorganize, highlight, and summarize key holdings according to specified style and formatting instructions.

### E.3 CASE STUDY

We provide more qualitative illustration of SSFO's benefits. As shown in Table 10 SSFO corrects parametric hallucinations made by the instruct model. In Table 11, example from FollowBench highlights SSFO's ability to retain strong instruction-following capabilities under complex, context-based scenarios. In Table 12 a cross-language case study underscores SSFO's capacity to maintain faithfulness even when queries and context are presented in a different language, such as Spanish.

Table 10: A case study showing how SSFO improves faithfulness on a NQ-swap dataset (Longpre et al., 2021). blue text indicates the correct, context-based answer; red text denote parametric-induced faithfulness hallucination.

| | |
|---|---|
| *Question* | Who plays grace in The Secret Life of the American Teenager? |
| *Context* | **Ferraro** (born July 24, 1986) is a Canadian actress and singer. **She is known for her portrayal of Grace Bowman in the television series The Secret Life of the American Teenager**. |
| *Base* | According to the text, Molly Ringwald plays the role of Grace Bowman in the television series The Secret Life of the American Teenager. **Hallucination** |
| *SSFO* | Based on the text, Ferraro plays the role of Grace Bowman in the television series The Secret Life of the American Teenager. **Context-Faithful** |

Table 11: Case study from FollowBench (Jiang et al., 2024): showing that SSFO retains strong instruction-following capabilities under a complex, context-based NLP task.

| | |
|---|---|
| *Instruction* | Below we show examples of soft drink. Following these examples, keep listing three more soft drink that satisfy following conditions: **1. not manufactured by coca cola company**; First Three Examples: |
| *Examples* | a) in chemistry, a phosphate is an anion, salt, functional group or ester derived from a phosphoric acid. b) coca cola, or coke, is a carbonated soft drink manufactured by the coca cola company.c) an egg cream is a cold beverage consisting of milk, carbonated water, and flavored syrup (typically chocolate or vanilla). |
| *ChatQA* | D) Fanta E) Pepsi Cola F) Mecca cola **Fanta is manufactured by coca cola company** |
| *SSFO* | Based on the given examples, here are three more soft drinks that satisfy the condition of not being manufactured by the Coca-Cola company: 1. Dr Pepper: a carbonated soft drink that is a blend of vanilla and cherry flavors. 2. Mountain Dew: a sweet and citrus-flavored carbonated soft drink that is owned by PepsiCo. 3. A&W Root Beer: a creamy and sweet carbonated soft drink that is flavored with vanilla and anise, owned by Dr Pepper Snapple Group (now Keurig Dr Pepper). **General Ability Retained** |

Table 12: Cross-language case study demonstrating that SSFO maintains faithfulness even when the query–answer pair is presented in Spanish (**ES**). **Blue** spans highlight correct, context-grounded information; **Red** spans mark parametric hallucinations.

| | |
|---|---|
| *Query* | **(EN)** Who first sent radio waves across the Atlantic? |
| | *(ES) ¿Quién fue el primero en enviar ondas de radio a través del Atlántico?* |
| *Context* | **(EN)** ...**Guglielmo Marconi** made his famous trans-Atlantic radio transmission for the first time in **1901**... (193 words omitted) |
| | *(ES) ...Cuando **Guglielmo Marconi** realizó por primera vez su famosa transmisión transatlántica de radio en **1901**...* |
| *Context-DPO* | **(EN)** According to the context, Guglielmo Marconi made the first-ever trans-Atlantic radio transmission in 1901.  **Faithful to context** |
| | *(ES) Según el contexto, no se puede determinar con certeza quién fue el primero en enviar ondas de radio a través del Atlántico, ya que la Corte Suprema de los Estados Unidos declaró que la decisión no tenía nada que ver con la reivindicación de Marconi...*  **Hallucination** |
| *SSFO* | **(EN)** Based on the context, Guglielmo Marconi made the first-ever trans-Atlantic radio transmission in 1901.  **Faithful to context** |
| | *(ES) Según el contexto, Guglielmo Marconi fue el primero en enviar ondas de radio a través del Atlántico, en 1901.*  **Faithful to context** |

# F LIMITATION

This paper introduces SSFO, a novel and efficient self-supervised method that significantly enhances faithfulness in RAG systems without inference overhead. While the current work demonstrates strong performance, one potential avenue for future investigation could be to extend its application and further assess its efficacy in more complex RAG scenarios, such as those involving extremely long contexts or the synthesis of information from multiple retrieved documents. Investigating the scalability and adaptability of SSFO to even more intricate and diverse knowledge domains represents a direction for continued advancement. SSFO is designed for regimes where the retrieved context is treated as authoritative. In scenarios with high noise or malicious retrieval (Wang et al., 2025; Huang et al., 2025), SSFO should be paired with upstream verification modules to prevent the faithful propagation of misinformation.

# G BROADER IMPACTS

Our work on Self-Supervised Faithfulness Optimization (SSFO) presents significant positive impacts for the advancement of reliable and trustworthy AI systems. By enhancing the faithfulness of Retrieval-Augmented Generation (RAG) models to provide context, SSFO alleviates the critical issue of faithfulness hallucination. This improvement leads to more accurate, verifiable, and dependable outputs from Large Language Models, which is crucial for applications where information integrity is paramount, such as in educational tools, scientific research, and systems providing critical information to the public. Reducing the propensity of LLMs to generate content that deviates from factual sources fosters greater user trust and promotes the responsible deployment of AI technologies in diverse real-world scenarios.

Furthermore, the self-supervised nature of SSFO offers considerable practical benefits that can accelerate the adoption of more faithful AI. By eliminating the need for costly human annotation or extensive, resource-intensive post-training procedures, our method makes the development of highly faithful models more accessible and economically viable for a broader range of researchers and developers. The negligible inference burden and the demonstrated strong generalization capabilities, including improved cross-lingual faithfulness and preservation of general instruction-following abilities, mean that the benefits of SSFO can be widely applied across different languages and tasks. This facilitates the development of more robust and equitable AI systems globally, contributing to a more informed and reliably assisted digital environment.

