# OpenReview forum: "SSFO: Self-Supervised Faithfulness Optimization for Retrieval-Augmented Generation"
_ICLR.cc/2026/Conference — ICLR 2026 Conference Withdrawn Submission_

### Official Review · Reviewer_Jc6W · 2025-10-28

**Soundness:** 2
**Presentation:** 3
**Contribution:** 3
**Rating:** 4
**Confidence:** 3

**Summary:**

This paper focuses on improving LLMs’ ability to follow context knowledge rather than relying on parametric knowledge. Based on DPO, the authors propose a self-supervised alignment approach, and further provide an analysis using likelihood displacement to explain why this method improves performance. They then introduce Shift-DPO to further enhance the results.

**Strengths:**

1. The proposed method outperforms the baselines without requiring any external ground-truth data.


2. The experiments cover multiple model families, demonstrating the generalization ability of the method.


3. The attribution of the performance gains to the benign application of likelihood displacement is insightful, and the corresponding analysis is reasonable.

**Weaknesses:**

1. Self-DPO is not a new technique, and similar approaches have already been explored. For example, [1] applies self-DPO to improve truthfulness.


2. The further proposed Shift-DPO is also introduced in prior work. Additionally, the paper claims that Shift-DPO “leads to a more pronounced suppression of the likelihood of the parametric response during optimization.” However, this seems inconsistent with the stated motivation of enabling better likelihood displacement, and the connection between the two is unclear. This point is therefore confusing.

3. Including results on additional model sizes would provide stronger evidence for the generalizability of the proposed method.


[1] Chen, Weixin, Dawn Song, and Bo Li. "GRATH: Gradual Self-Truthifying for Large Language Models." Forty-first International Conference on Machine Learning.

**Questions:**

See Weaknesses

---

> ### Author Response · Authors · 2025-11-20
> **Response to Reviewer Jc6W (Part I)**
>
> Thanks for your time and the valuable comments. We appreciate your recognition of our comprehensive experiments across model families and, in particular, your acknowledgment that our analysis of likelihood displacement is insightful and reasonable. Below, we provide detailed responses to each of your comments and hope to address any further considerations you may have.
>
> **W1: Contributions of SSFO in Addressing Contextual Faithfulness.**
> > Self-DPO is not a new technique, and similar approaches have already been explored. For example, [1] applies self-DPO to improve truthfulness.
>
> We thank the reviewer for their insightful feedback and for highlighting the related work on self-preference alignment, such as GRATH.
> We wish to respectfully clarify the distinctions that separate our work from these existing approaches:
>
> 1. **Different Problem Domain**: Faithfulness vs. Truthfulness. Our work focuses specifically on faithfulness hallucination. This is the challenge of forcing a model to adhere strictly to a provided, external context, even when that context conflicts with the model's internal parametric knowledge. This is distinct from the problem of "truthfulness" (tackled by GRATH), which typically aims to improve the factual accuracy of a model's parametric knowledge (i.e., its knowledge of world facts).
>
> 2. **Analysis & Mechanism: Benign Likelihood Displacement.** A major contribution of our paper is the analysis of why this specific contrast works. We are the first to demonstrate that this alignment process operates via a "benign likelihood displacement". Our analysis shows that SSFO does not just learn a vague preference; it actively shifts probability mass away from parametric-based tokens ($z_p$) and towards context-aligned tokens ($z_c$).
>
> 3. **Method Variant (SSFO-$\lambda$)**: This analysis directly led to our second contribution, SSFO-$\lambda$. Because we identify this mechanism, we are able to propose a variant that intentionally encourages and amplifies this beneficial displacement. This is a counterintuitive  finding, as most prior DPO literature treats likelihood displacement as a problem to be mitigated. Our work is the first to re-frame it as a desirable feature to be exploited for context faithfulness.
>
>
>
> **W2: Clarification of the Role between Parametric Suppression and Likelihood Displacement**
> > The further proposed Shift-DPO is also introduced in prior work. Additionally, the paper claims that Shift-DPO “leads to a more pronounced suppression of the likelihood of the parametric response during optimization.” However, this seems inconsistent with the stated motivation of enabling better likelihood displacement, and the connection between the two is unclear. This point is therefore confusing.
>
> We sincerely thank the reviewer for this insightful question. We acknowledge that the connection between “suppressing the parametric response” and “enabling better likelihood displacement” was not fully explicated in the main text. We clarify the causal link below:
>
>
>
> 1. **Why stronger suppression of the parametric response achieves benign likelihood displacement:** In our setting, the "preferred" response $y_c'$ is a "silver" example generated by the model itself, and it still contains unwanted parametric knowledge. Our goal is not to preserve the absolute likelihood of this particular sequence, but to reallocate probability mass from parametric-based tokens to context-grounded tokens $z_c$.
>
>     Introducing $\lambda > 1$ applies a stronger penalty to the parametric-only response $y_p$, **which in turn shifts probability mass away from all parametric-based tokens (in both $y_p$ and $y_c'$) and toward context-aligned tokens.** This is precisely the “benign likelihood displacement” we analyze in Section 3.3: the likelihood of the two specific sequences $y'_c$ and $y_p$ can decrease, while the model becomes more faithful by increasing the likelihood of context-grounded response overall.
>
> 2. **Effect of λ on likelihood displacement:** Following your suggestion, we have added a derivation in Appendix C.1 showing that setting $\lambda > 1$ in our SSFO-λ objective amplifies likelihood displacement compared to $\lambda = 1$. Concretely, both the log-likelihood of the preferred response and that of the parametric response decrease, while the preference margin between them increases.
>
>
> 3. **Relation to prior work:** Prior Work (e.g., DPO-Shift, AlphaPO): typically view likelihood displacement as an **undesirable side effect of DPO**. Their "preferred" response $y_w$ is typically a high-quality, "golden" example (e.g., human-written). Reducing its likelihood is harmful, so they aim to alleviate this displacement. In contrast, our $y_c'$ is imperfect. We want to displace probability mass away from the parametric components within $y_c'$. Therefore, encouraging this displacement via $\lambda > 1$ serves our goal, suppressing the shared parametric hallucinations while preserving the context-grounded tokens.

---

> ### Author Response · Authors · 2025-11-20
> **Response to Reviewer Jc6W (Part II)**
>
> **Response to Q3: Extended Evaluation on Generalizability**
> > Including results on additional model sizes would provide stronger evidence for the generalizability of the proposed method.
>
> We appreciate the reviewer’s suggestion to evaluate additional model sizes. We have included the results in Table 8 in our original submission, with a hyperlink provided below Table 1 for convenience. To assess the scalability of SSFO, we apply it to Qwen2.5 models ranging from 1.5B to 72B parameters and report relative gains over each instruct baseline on five benchmarks. Across all model sizes, SSFO consistently improves robustness, yielding +3% to +14% span-EM gains, and enhances response quality, boosting closed-book QA and long-form generation metrics by up to +6%.
>
>
> | Instruct Model   | Method   | NQ-Swap (Span EM) | Memo-Trap (Span EM) | NQ-Open (Span EM) | SQUAD (Span EM) | Eli5 (R-1 F1) | Eli5 (R-2 F1) | Eli5 (RL F1) |
> | :--------------- | :------- | :---------------- | :------------------ | :---------------- | :-------------- | :------------ | :------------ | :----------- |
> | **Qwen2.5 1.5B** | Baseline | 25.44%            | 65.78%              | 79.17%            | 88.80%          | 22.11%        | 4.26%         | 19.40%       |
> |                  | **SSFO** | **41.12%**        | **79.65%**          | **83.88%**        | **92.90%**      | **28.10%**    | **8.86%**     | **24.87%**   |
> | **Qwen2.5 3B**   | Baseline | 76.38%            | 47.66%              | 82.11%            | 59.12%          | 23.57%        | 6.60%         | 21.11%       |
> |                  | **SSFO** | **88.20%**        | **76.95%**          | **81.32%**        | **92.40%**      | **25.74%**    | **9.16%**     | **23.31%**   |
> | **Qwen2.5 7B**   | Baseline | 79.35%            | 54.19%              | 82.29%            | 90.30%          | 23.08%        | 6.06%         | 20.55%       |
> |                  | **SSFO** | **84.18%**        | **57.66%**          | **83.88%**        | **92.00%**      | **24.63%**    | **7.17%**     | **21.83%**   |
> | **Qwen2.5 14B**  | Baseline | 82.15%            | 64.08%              | 82.49%            | 90.00%          | 22.48%        | 5.62%         | 20.03%       |
> |                  | **SSFO** | **85.04%**        | **66.52%**          | **84.14%**        | **92.80%**      | **25.42%**    | **8.19%**     | **22.97%**   |
> | **Qwen2.5 72B**  | Baseline | 81.84%            | 66.99%              | 83.50%            | 91.20%          | 21.85%        | 5.26%         | 19.45%       |
> |                  | **SSFO** | **87.51%**        | **67.39%**          | **84.48%**        | **92.70%**      | **22.46%**    | **5.71%**     | **19.96%**   |

---

### Official Review · Reviewer_iQ9y · 2025-10-31

**Soundness:** 2
**Presentation:** 3
**Contribution:** 2
**Rating:** 2
**Confidence:** 4

**Summary:**

This paper introduces Self-Supervised Faithfulness Optimization (SSFO), a RAG alignment method that creates a preference dataset by contrasting a model's own context-based generation (preferred) against its parametric-only generation (dispreferred). The model is then aligned using DPO. The authors posit this induces a "benign likelihood displacement," shifting probability from parametric tokens to context-aligned tokens and introduce the hyper-parameter to encourage this process. The paper claims state-of-the-art faithfulness, strong robustness to conflicting knowledge (e.g., NQ-Swap, MemoTrap), good generalization, and high data efficiency from only hundreds of training examples.

**Strengths:**

- The method's primary strength is its simplicity. It avoids costly human or teacher annotation, and achieving significant gains from <1000 self-generated examples is a notable practical contribution.
- The "benign likelihood displacement" concept provides a comfirmation of the method prioritizes contextual information over parametric memory.
- The method demonstrates empirical gains on benchmarks specifically designed to test robustness against conflicting internal knowledge (NQ-Swap, MemoTrap). It also shows that SSFO generalizes well cross-lingually and preserves general instruction-following capabilities.

**Weaknesses:**

- The entire SSFO framework is built on the critical, unstated assumption that the retrieved context is always the source of truth. The method explicitly trains the model to suppress its (potentially correct) internal knowledge in favor of any provided context. This is a major blind spot. In realistic RAG scenarios involving noisy, irrelevant, or factually incorrect context, SSFO would likely amplify this critical failure mode, forcing the model to "faithfully" repeat misinformation. The paper fails to evaluate this obvious scenario or acknowledge this foundational weakness. There is an entire line of works on dynamic contextual faithfulness ([Huang et al., ICLR 2025](https://openreview.net/forum?id=K2jOacHUlO), [Wang et al., ACL 2025](https://aclanthology.org/2025.acl-long.1476/)), which is ignored in the paper.
- The paper omits the exact prompt template used for the "Instruct Model" baseline. This is a crucial omission, given that SSFO's goal is to force strict adherence to the context, it is plausible that its entire training effect could be replicated by a well-engineered prompt (e.g., "Using only the provided context, answer the following..."). The authors do not provide evidence that SSFO achieves an optimization benefit beyond what simple prompting could accomplish, which would significantly weaken the paper's contribution.
- The empirical evaluation against other baselines is also undermined by several issues. Several baseline results are abnormally low (e.g., SCOPE on Mistral-7B, CAD on Llama-3-8B) without explanation, casting doubt on the implementation's correctness and, consequently, the validity of SSFO's superiority. Although the self-training protocol is a core proposal, the fact that the baselines are trained on different data makes the comparison even harder to interpret.

**Questions:**

- There are widespread citation format errors (e.g., inline parenthetical `(2020)` instead of `\citep{...}`).
- Typo in Table 3: "Crose-language".
- Formatting in Table 1: The "Human/Superior AI Supervision" column has excessive horizontal space, making the other content too small to read.

---

> ### Author Response · Authors · 2025-11-20
> **Response to Reviewer iQ9y (Part I)**
>
> We sincerely appreciate your critical and insightful review. We appreciate your recognition of SSFO’s simplicity and practical value. Below, we provide detailed responses to each of your concerns about the reliance on context correctness, comparison to prompting-only baselines, and baseline implementations/data alignment, and hope to address any further considerations you may have.
>
>
> **W1: Clarification on the “Context-as-Truth” Assumption, Robustness to Noisy Contexts.**
> > The entire SSFO framework is built on the critical, unstated assumption that the retrieved context is always the source of truth. The method explicitly trains the model to suppress its (potentially correct) internal knowledge in favor of any provided context. This is a major blind spot. In realistic RAG scenarios involving noisy, irrelevant, or factually incorrect context, SSFO would likely amplify this critical failure mode, forcing the model to "faithfully" repeat misinformation. The paper fails to evaluate this obvious scenario or acknowledge this foundational weakness. There is an entire line of works on dynamic contextual faithfulness (Huang et al., ICLR 2025, Wang et al., ACL 2025), which is ignored in the paper.
>
>
> We thank the reviewer for this insightful observation. We have incorporated these works into our revised Limitations sections to better contextualize our contribution. Below, we respectfully clarify the scope and necessity of SSFO:
>
>
> 1. **SSFO can be naturally composed with upstream source-verification or dynamic-faithfulness modules**, such as methods that verify document reliability or contextualize retrieved evidence (e.g., https://openreview.net/forum?id=JnWJbrnaUE, https://arxiv.org/pdf/2311.08377). In such a pipeline, the verifier first identifies trustworthy context, and SSFO is then applied to ensure that, conditional on the accepted context, the model replies faithfully.
>
> 2. **SSFO is beneficial for a large and important subset of applications in which the retrieved context is treated as the authoritative source of truth**. Typical scenarios include:
>
>       - **Private/Vertical Knowledge (Legal/Medical):** The model’s pre-trained weights are unaware of private case files or patient records. The retrieved context is the only valid source of information.
>
>       - **Time-Sensitive/Dynamic Knowledge**: In Real-Time QA (e.g., TempRAG), the retrieval layer provides the latest news. The model's internal knowledge is outdated; therefore, strict adherence is required for correctness.
>
>       - **Extraction & Summarization**: When a user asks a model to "summarize this meeting transcript" or "extract entities from this contract," the model must be faithful to the input, even if the input contains factual errors regarding the outside world.
>
>      In such settings, mitigating misinformation is the responsibility of the retrieval layer. Therefore, our evaluations (e.g., MemoTrap, NQ-Swap) focus specifically on resolving conflicts between context and internal knowledge, rather than handling noisy retrieval.

---

> ### Author Response · Authors · 2025-11-20
> **Response to Reviewer iQ9y (Part II)**
>
> **W2: Disclosure of Prompt Templates and Comparative Analysis against Strong Prompting Baselines.**
> > The paper omits the exact prompt template used for the "Instruct Model" baseline. This is a crucial omission, given that SSFO's goal is to force strict adherence to the context, it is plausible that its entire training effect could be replicated by a well-engineered prompt (e.g., "Using only the provided context, answer the following..."). The authors do not provide evidence that SSFO achieves an optimization benefit beyond what simple prompting could accomplish, which would significantly weaken the paper's contribution.
>
>
> We thank the reviewer for highlighting the missing prompt specification and apologize for the oversight. In the revised version, we have included the exact prompt templates used for all *Instruct Model* baselines. Our code repository is publicly available, ensuring complete reproducibility.
>
> In our original setup, the instruction already asked the model to answer based on the retrieved passage:
>
> **Original Prompt Template**
>
> > Based on the following context: `{context}`
> > Answer the following question: `{question}`
>
> To address the reviewer’s concern more directly, we further evaluated stronger “strict” prompting baselines that explicitly prohibit using parametric knowledge:
>
> **Strict Prompt**
>
> > You are given a question and a context passage retrieved for this question.
> > **Context:** `{context}`
> > You must answer **only** using information from the context. Answer the following **Question:** `{question}`.
>
> We also tested a **context-faithful prompt** adapted from prior work on faithful generation (https://aclanthology.org/2023.findings-emnlp.968.pdf), which emphasizes answering strictly from the given text:
>
> **Context-Faithful Prompt**
>
> > **Prompt instruction:** Read the given information and answer the corresponding question.
> > **Instruction:** Answer the question based on the provided input–output pairs.
> > **Example:** Bob said `{context}`.
> > Q: `{question}` in Bob’s opinion based on the given text?
>
> Across these prompt variants, we consistently observe that SSFO yields additional gains in context faithfulness and answer quality beyond what prompt engineering alone can achieve. While prompts operate only at inference time and may be only partially followed by the model, SSFO modifies the training objective to systematically down-weight parametric completions that conflict with the context and up-weight context-aligned tokens. This optimization effect cannot be replicated by natural-language prompting alone.
>
> | Model / Prompt Type      | NQ-Swap | NQ-Open | Memotrap |
> | ------------------------ | ------- | ------- | -------- |
> | **llama3-3-8B-Instruct** |         |         |          |
> | Original Prompt          | 73.54%  | 80.15%  | 73.60%   |
> | Strict Prompt            | 74.55%  | 80.04%  | 74.78%   |
> | Context-faithful Prompt  | 74.57%  | 80.11%  | —        |
> | **Qwen2.5-7B-Instruct**  |         |         |          |
> | Original Prompt          | 79.35%  | 82.29%  | 54.19%   |
> | Strict Prompt            | 80.13%  | 82.60%  | 55.73%   |
> | Context-faithful Prompt  | 79.62%  | 83.26%  | —        |
> | **llama3-3-8B-SSFO**     |         |         |          |
> | Original Prompt          | 81.23%  | 84.40%  | 76.28%   |
> | Strict Prompt            | 82.01%  | 83.36%  | 76.76%   |
> | Context-faithful Prompt  | 80.38%  | 82.22%  | —        |
> | **Qwen2.5-7B-SSFO**      |         |         |          |
> | Original Prompt          | 84.18%  | 83.88%  | 57.66%   |
> | Strict Prompt            | 84.64%  | 84.14%  | 59.00%   |
> | Context-faithful Prompt  | 84.39%  | 85.84%  | —        |
>
> MemoTrap is excluded from the Context-Faithful Prompt evaluation due to format incompatibility.

---

> ### Author Response · Authors · 2025-11-20
> **Response to Reviewer iQ9y (Part III)**
>
> **W3: Verification of Baseline Implementations and Explanations for Performance Variances.**
> > The empirical evaluation against other baselines is also undermined by several issues. Several baseline results are abnormally low (e.g., SCOPE on Mistral-7B, CAD on Llama-3-8B) without explanation, casting doubt on the implementation's correctness and, consequently, the validity of SSFO's superiority. Although the self-training protocol is a core proposal, the fact that the baselines are trained on different data makes the comparison even harder to interpret.
>
> We thank the reviewer for pointing out the performance issue.
>
>
> 1. For CAD on Llama-3-8B: This behavior stems from an inherent trade-off in decoding strategies: while the standard penalty hyperparameter ($\alpha=0.5$, as recommended in Shi et al., 2024 https://aclanthology.org/2024.naacl-short.69/) is effective in conflict scenarios like NQ-Swap (74.83% vs 73.54% baseline) and MemoTrap. It inadvertently degrades the fluency and coherence of the generation for standard QA tasks, causing a drop in NQ-Open, SQuAD performance. To ensure our evaluation is as robust as possible, we followed your feedback and conducted a grid search over $\alpha \in [0.1, 1.0]$ for the Llama-3 family. While optimizing $\alpha$ yielded improved results for CAD on standard QA tasks, SSFO still consistently outperforms the optimized CAD baseline across all metrics. For instance, even with the optimal $\alpha$, CAD achieves 81.44% on NQ-Open, whereas SSFO achieves 84.40%. We have updated Table 1 to reflect these "best-case" baseline results, strengthening the validity of our comparative analysis.
>
> 2. For SCOPE on Mistral-7B: We utilized the official source code (https://github.com/sngdng/scope-faithfulness). SCOPE synthesizes negative examples via a "token-level mixture" of a fine-tuned model and a pre-trained model. We observed that Mistral-7B is sensitive to this injection. Unlike Llama-3, Mistral-7B frequently generates incoherent or repetitive text when trained on these noisy synthetic negatives, leading to the low scores reported.
>
>
> **W4: Clarification on Experimental Settings.**
>
> Regarding the concern that baselines are trained on different data, we clarify our comparison protocol as follows.
>
> For *decoding-only* baselines (CAD, DECORE), no additional training is performed. We apply the authors’ decoding algorithms on top of the *same* base instruct models (Llama-3-8B, Qwen-2.5-7B, Mistral-7B), under the *same* RAG pipeline and evaluation datasets as SSFO (see Table 1). Thus, any performance differences here are due to the decoding method itself rather than differences in training data.
>
> For *supervised post-training* baselines (Trust-Align, ChatQA, Context-DPO):
> 1. Their core proposition is the use of high-quality, supervision-heavy data (GPT-4 synthesized or human-annotated). To respect the original design of these methods, we use the official open-source model weights released by the authors, rather than retraining them on our own data. SSFO achieves comparable or superior performance using only ~800 self-generated samples, compared to their 5k–30k expensive external samples (Table 3). This is because SSFO preserves the model’s native style for both preferred and dispreferred answers, enabling optimization to focus purely on faithfulness improvements without introducing a confounding style gap.
>
> 2. We do not construct any training data targeted to our test sets. SSFO was trained only on the generic MS-MARCO dataset and evaluated in a zero-shot manner across **eight** diverse out-of-domain benchmarks (NQ, SQuAD, MemoTrap, etc.). This setup avoids benchmark-specific tuning and ensures a fair and comparable evaluation regime across all methods.
>
>
> **W5: Formatting, Typos, and Citation Style.**
>
> We thank the reviewer for the careful proofreading and have addressed all listed issues in the revision. Specifically, we have standardized citation formats throughout the paper, corrected the "Crose-language" typo in Table 3, and adjusted the column widths in Table 1 to reduce excessive whitespace and improve readability.

---

### Official Review · Reviewer_zPuA · 2025-11-01

**Soundness:** 3
**Presentation:** 3
**Contribution:** 3
**Rating:** 6
**Confidence:** 3

**Summary:**

This paper proposes Self-Supervised Faithfulness Optimization which is a post-training method for RAG that builds preference pairs from the same model by contrasting answers generated with retrieved context against answers generated without context. The pairs are optimized with DPO, and analyzed empirically supported mechanism is "benign likelihood displacement", shifting probability mass from parametric tokens to context aligned tokens.
The authors also propose a simple variant of SSFO-Lamda which unweights the pressure to suppress the parametric response in the DPO objective. Across different models (llama, qwen and mistral), both the proposed methods show improvement in Span-EM and long form metrics, with no extra inference time costs and using only hundreds of self generated pairs.

**Strengths:**

1) The overall idea is simple, yet gives a strong empirical payoff, when doing self supervised the preference pairs are easy to generate and avoid costly human labels, this directly aligns to a context adherence principle. This reduces supervision and avoids extra inference burden.
2) The paper explains why encouraging displacement can be beneficial when preferred examples are silver, then introduces SSFO-lambda that rescores the DPO objective so the gradient puts stronger negative weight, which is a strong motivation with has a broad application in the current field.
3)  ​Results in Table 1 shows how SSFO-lamda outperforms strong decoding strategies and post training baselines on both robustness and quality response for diverse models (llama and qwen)
4) The authors also show the data efficiency of the proposed method, and how only 400-500 pairs can achieve upto 85% of the total gains over the instruct baselines.

**Weaknesses:**

1) Results depend on retrieval quality, and the paper states a “standard RAG prompt” and datasets, but does not detail the retriever configuration or ablations to retrieval quality.
2) The authors report LFS which uses GPT-4 with a provided prompt, while standard, it introduces judge bias and lacks calibration against human labels or alternative factuality metrics, and there is no reliability analysis

**Questions:**

1) The paper compares against key faithfulness methods but omits other displacement-aware alignment baselines like DPO-shift and AlphaPO in experiments even though they are mentioned, adding these results in the table could strengthen their claims.
2) Consider adding RAGTruth which is cited in the paper, to show faithfulness under human-curated hallucination settings.
3) The paper misses an explicit evaluation under retrieval noise (distractor setting), missing or contradictory contexts, long-context settings, such experiments could strengthen the claims of the paper.

---

> ### Author Response · Authors · 2025-11-20
> **Response to Reviewer zPuA (Part I)**
>
> Thank you for your positive assessment and insightful comments. We are encouraged that you recognize the simplicity and strong empirical payoff of our proposed method, as well as its advantage in avoiding costly human labels through self-supervision. Below, we provide detailed responses to each of your concerns and hope to address any further considerations you may have.
>
> **W1: Retriever Configuration and Dependence on Retrieval Quality.**
> > Results depend on retrieval quality, and the paper states a “standard RAG prompt” and datasets, but does not detail the retriever configuration or ablations to retrieval quality.
>
> We thank the reviewer for raising the concern about retrieval quality.
> 1. In all experiments, we use the publicly available MS MARCO passage-ranking outputs as the source of candidate contexts, and we will made this configuration explicit in the Implementation Details section of the revised paper.
> 2. To ensure the quality of the preference pairs without relying on external supervision, we employ a lightweight self-filtering step using the model's native capabilities (described in Appendix D.3). Specifically, we prompt the model with the instruction: "If you are not sure of the answer, please reply 'I don't know'." We then discard any instance where the model refuses to answer based on the retrieved context. This effectively removes question–passage pairs that are clearly uninformative, controlling for obvious retrieval failures before SSFO training begins.
> 3. To directly assess the dependence on retrieval quality, we have added an ablation over three retrieval conditions:
>
>    - **Golden retrieval:** the context consists only of the gold passage that contains the answer.
>    - **Noisy retrieval:** the gold passage is mixed with retrieved non-relevant passages from MS MARCO.
>    - **Reject retrieval:** the retrieved context does not contain the answer.
>
>    | Model / Regime          | NQ-Swap | NQ-Open | MemoTrap |
>    | ----------------------- | ------: | ------: | -------: |
>    | **LLaMA3-8B-Instruct**  |         |         |          |
>    | Golden retrieval        |   85.05 |   84.39 |    55.07 |
>    | Noisy retrieval         |   84.18 |   83.88 |    57.66 |
>    | Reject retrieval        |   82.95 |   80.62 |    51.24 |
>    | **Qwen2.5-7B-Instruct** |         |         |          |
>    | Golden retrieval        |   80.53 |   85.04 |    77.86 |
>    | Noisy retrieval         |   81.23 |   84.40 |    76.28 |
>    | Reject retrieval        |   77.59 |   80.04 |    74.27 |
>
>    Across both models and all three datasets, moving from golden to reject retrieval degrades performance by at most about 2–5 points, and the noisy regime performs very close to (and in some cases slightly better than) golden retrieval. This indicates that SSFO is relatively robust to retrieval noise and does not rely on unrealistically perfect retrieval.

---

> ### Author Response · Authors · 2025-11-20
> **Response to Reviewer zPuA (Part II)**
>
> **W2: On the Validity of GPT-4 Evaluation**
> > The authors report LFS which uses GPT-4 with a provided prompt, while standard, it introduces judge bias and lacks calibration against human labels or alternative factuality metrics, and there is no reliability analysis
>
> We thank the reviewer for raising the concern about judge bias in LFS. For most datasets in our evaluation, we rely on standard, objective metrics (e.g., exact match for QA-style tasks). We only use the GPT-4–based LFS metric for long-form generation, where purely lexical metrics such as ROUGE are known to be biased and often misaligned with human judgments of factuality. In these cases, LFS is used as a *complementary* signal rather than the sole basis for our conclusions.
>
> Importantly, the same GPT-4 judge and the same prompt are applied uniformly to all systems, so any residual bias in the judge is shared across methods and does not preferentially benefit SSFO.
>
> To calibrate LFS against human judgments, we conducted a reliability study on a random subset of \(N = 50\) long-form outputs, labeled independently by a human annotator and GPT-4 into three categories: *Faithful*, *Partially Faithful*, and *Unfaithful*.
>
> |                               | GPT-4: Faithful | GPT-4: Partially Faithful | GPT-4: Unfaithful | Human Total |
> | ----------------------------- | --------------- | ------------------------- | ----------------- | ----------- |
> | **Human: Faithful**           | 22              | 3                         | 0                 | 25          |
> | **Human: Partially Faithful** | 2               | 12                        | 1                 | 15          |
> | **Human: Unfaithful**         | 0               | 1                         | 9                 | 10          |
> | **GPT-4 Total**               | 24              | 16                        | 10                | 50          |
>
> Overall, GPT-4 agrees exactly with the human labels in 43 out of 50 cases (\(86\%\) agreement). Most discrepancies occur between *Faithful* and *Partially Faithful*, while cases labeled *Unfaithful* by humans are never judged as *Faithful* by GPT-4. This pattern suggests that GPT-4 is conservative with respect to strong faithfulness, and errors are mostly near the decision boundary rather than misjudgments.

---

### Official Review · Reviewer_apKU · 2025-11-06

**Soundness:** 3
**Presentation:** 3
**Contribution:** 3
**Rating:** 6
**Confidence:** 3

**Summary:**

The paper introduces SSFO, a post‑training alignment method for RAG that creates preference pairs without human or stronger‑LLM supervision. For cheaper training, SSFO constructs preference data pairs by contrasting the model’s outputs generated with context versus without context. The model is then optimized with a DPO‑style objective that widens the margin between these two responses. They show that their methodology improves the performance on several short‑form QA datasets (NQ‑Swap, MemoTrap, NQ‑Open, SQuAD) and long‑form metrics on ELI5/WikiPassageQA, preserves instruction following (FollowBench), and shows cross‑lingual generalization (XQuAD‑ES, DuReader‑ZH), often with only hundreds of self‑generated training pairs.

**Strengths:**

- They propose a self-supervised and efficient training method. Preference pairs are self‑generated, and only hundreds of examples suffice.
- They conduct a systemic evaluation with several benchmarks from various aspects, including Robustness, Response Quality, Cross-language Response Quality, and Instruction Following Ability.
- They prove that their method is effective across LLMs.
- They not only propose the effective training method but also try to explain why their method works through benign likelihood displacement.

**Weaknesses:**

Actually, I don't see many weaknesses in this paper. However, one question is about how the authors ensure that the no-context generations are indeed dispreferred responses. In some cases, no-context generations might still be factually correct and meaningfully not different from the context-grounded response. The paper could be clearer about how it guarantees that these no-context responses are dispreferred and meaningfully distinct from context-grounded responses.

Moreover, they use GPT-4 as LLM-as-Judge to calculate LFS. This might introduce some LLM bias during the evaluation. Did you check some responses manually to verify the validity and consistency of GPT-4?

It's minor, but for readability, please use "\citep" for citations.

**Questions:**

Please see the weaknesses.

---

> ### Author Response · Authors · 2025-11-20
> **Response to Reviewer apKU (Part I)**
>
> Thanks for your positive review and constructive comments. We appreciate your recognition of our method's self-supervised efficiency, the systematic evaluation across diverse aspects, and our theoretical analysis of benign likelihood displacement. In the response below, we provide clarifications on how we ensure the distinctiveness of dispreferred responses and discuss the manual verification we conducted to mitigate LLM-as-Judge bias.
>
>
> **W1: Clarification on Negative Sample Construction**
> > One question is about how the authors ensure that the no-context generations are indeed dispreferred responses. In some cases, no-context generations might still be factually correct and meaningfully not different from the context-grounded response. The paper could be clearer about how it guarantees that these no-context responses are dispreferred and meaningfully distinct from context-grounded responses.
>
> We thank the reviewer for raising this important point.
>
> **1. What we assume about no-context responses**: Our method does not assume that the dispreferred no-context response ($y_p$) is factually wrong. Instead, the "preference" in SSFO is defined strictly along the axis of contextual adherence: We label the no-context response ($y_p$) as "dispreferred" because it relies on parametric memory, whereas the preferred context-grounded response ($y_c'$) utilizes the retrieved evidence. **We only rely on the weak, aggregate assumption that preferred response $y_c'$ is more faithful to the specific provided context than dispreferred response $y_p$.** This effectively trains the model to prioritize external information over internal memory.
>
> **2. Why meaningfully similar no-context responses are not a problem**: Even when the dispreferred no-context response happens to be very close to the context-grounded answer, this does not harm training. The DPO objective naturally down-weights such pairs. Specifically, when preferred and dispreferred responses are identical or nearly so, their log-likelihood ratio is close to zero. **Consequently, the gradient contribution from these specific pairs becomes negligible.**
>
>
> **W2: On the Validity of GPT-4 Evaluation**
> > They use GPT-4 as LLM-as-Judge to calculate LFS. This might introduce some LLM bias during the evaluation.
>
> We thank the reviewer for raising the concern about judge bias in LFS. For most datasets in our evaluation, we rely on standard, objective metrics (e.g., exact match for QA-style tasks). We only use the GPT-4–based LFS metric for long-form generation, where purely lexical metrics such as ROUGE are known to be biased and often misaligned with human judgments of factuality. In these cases, LFS is used as a *complementary* signal rather than the sole basis for our conclusions.
>
> Importantly, the same GPT-4 judge and the same prompt are applied uniformly to all systems, so any residual bias in the judge is shared across methods and does not preferentially benefit SSFO.
>
> To calibrate LFS against human judgments, we conducted a reliability study on a random subset of \(N = 50\) long-form outputs, labeled independently by a human annotator and GPT-4 into three categories: *Faithful*, *Partially Faithful*, and *Unfaithful*.
>
> |                               | GPT-4: Faithful | GPT-4: Partially Faithful | GPT-4: Unfaithful | Human Total |
> | ----------------------------- | --------------- | ------------------------- | ----------------- | ----------- |
> | **Human: Faithful**           | 22              | 3                         | 0                 | 25          |
> | **Human: Partially Faithful** | 2               | 12                        | 1                 | 15          |
> | **Human: Unfaithful**         | 0               | 1                         | 9                 | 10          |
> | **GPT-4 Total**               | 24              | 16                        | 10                | 50          |
>
> Overall, GPT-4 agrees exactly with the human labels in 43 out of 50 cases (\(86\%\) agreement). Most discrepancies occur between *Faithful* and *Partially Faithful*, while cases labeled *Unfaithful* by humans are never judged as *Faithful* by GPT-4. This pattern suggests that GPT-4 is conservative with respect to strong faithfulness, and errors are mostly near the decision boundary rather than misjudgments.
>
> **We thank the reviewer for the careful proofreading. We have standardized citation formats throughout the paper.**

---

### Note · Authors · 2025-12-02

I have read and agree with the venue's withdrawal policy on behalf of myself and my co-authors.